# From Predictors to Samplers via the Training Trajectory

**Soumya Ram**[1]***Akhila Ram**[2]
[1] Independent Researcher
[2] Department of Computer Science, Massachusetts Institute of Technology

## Abstract

Sampling from trained predictors is fundamental for interpretability and as a compute-light alternative to diffusion models, but local samplers struggle on the rugged, high-frequency functions such models learn. We observe that standard neural-network training implicitly produces a coarse-to-fine sequence of models. Early checkpoints suppress high-degree/ high-frequency components (Boolean monomials; spherical harmonics under NTK), while later checkpoints restore detail. We exploit this by running a simple annealed sampler across the training trajectory, using early checkpoints for high-mobility proposals and later ones for refinement. In the Boolean domain, this can turn the exponential bottleneck arising from rugged landscapes or needle gadgets into a near-linear one. In the continuous domain, under the NTK regime, this corresponds to smoothing under the NTK kernel. Requiring no additional compute, our method shows strong empirical gains across a variety of synthetic and real-world tasks, including constrained sampling tasks that diffusion models are unable to handle.

## 1 Introduction

In contrast to the trend toward billion-parameter Transformer LLMs, model deployments for medicine, recommendation systems, and decision support based on structured data continue to be dominated by small CNNs/MLPs. These dominate AI in medical devices (Singh et al., 2025; Mienye et al., 2025), production models for personalized recommendations (Feng et al., 2024), and decision support models for credit scoring, recidivism risk, insurance underwriting, and hospital operations triage (eba, 2023; Grinsztajn et al., 2022; McElfresh et al., 2023; Holzmüller et al., 2024).

Despite their relative simplicity, they can be opaque and encode brittle shortcuts. For example, a dermatology CNN approved for EU clinical use was shown post-hoc to over-weight surgical skin markings/rulers rather than lesion content; adding a simple violet marker to the same benign lesion skyrocketed its melanoma probability (Winkler et al., 2019; Bevan & Atapour-Abarghouei, 2022). Sampling minimal counterfactual edits could have revealed this shortcut. Such cases underscore the importance of sampling from the trained predictor.

Apart from interpretability, we often want to sample from a trained predictor to pick high-value candidates for active learning. For instance, many works sample fit DNA sequences from models predicting DNA-transcription factor (TF) affinity (de Almeida et al., 2022; Reddy et al., 2024). This works well as DNA-TF affinity assays test millions of sequences per experiment, often from largely random libraries—enabling near-unbiased exploration of sequence space (Gallego Romero & Lea, 2023). Similarly, for protein engineering, many methods sample sequences from a learned fitness model for bayesian optimization (Hu et al., 2022; Ren et al., 2022).

However, these sampling tasks can become difficult for certain common landscapes. When the landscape is rugged, high-frequency, high-magnitude fluctuations create many sharp local optima. Another key culprit is synergy — outcomes depend on rare combinations of variables—so the individual effects look innocuous while the rare, joint effect is large. Such concealed interactions evade single-step proposals and short-horizon heuristics, which only "see" myopic gains. As a result, discovering the right multi-variable change becomes essentially a needle-in-a-haystack problem, with

---

*Corresponding author: soumya1910@gmail.com

success of random exploration drops exponentially as the number and order of synergistic interactions grow, along with the number of spurious variables.

To deal with this difficulty, one option is to train a reward-conditioned diffusion or discrete-walk jump sampler. These are powerful generative approaches, but in the settings we focus on they have three drawbacks: (1) They require training a separate generative model in addition to the predictor, which can demand substantial extra compute compared to reusing an already-trained predictor with test-time trajectory-annealed MCMC. In many domains, strong predictors have already been trained on large, unbiased data, and practitioners operate in a compute-constrained regime where training an additional generative model is not feasible. (2) Implementing hard constraints such as a Hamming-radius ball or minimal counterfactual edits typically requires additional machinery (e.g., auxiliary guidance networks Shen et al. (2024) or SMC-style schemes Wu et al. (2023)), rather than a simple modification of the sampling rule. (3) They do not directly support sampling from a deployed model for interpretability, whereas our method operates on the existing predictor without any additional training.

We study the plug-and-play test-time sampling problem for a trained scalar predictor $f^* : \mathcal{X} \to \mathbb{R}$, and we aim to draw from the Gibbs density it induces, $\pi^*(x) \propto \exp\{f^*(x)\}$, optionally under hard constraints. Our solution is *trajectory annealing*: rather than sample only from $\pi^*$, we traverse training checkpoints $\{f_t\}_{t=0}^T$ and run brief MCMC updates targeting $\pi_t(x) \propto \exp\{f_t(x)\}$ before arriving at $\pi^*$.

This exploits the coarse-to-fine learning dynamics of predictors: early checkpoints suppress high-degree components, smoothing the landscape for rapid mixing. This yields two wins. For $\pi^*(x)$ with high-magnitude, high-frequency variation, early checkpoints bypass the rugged barriers that cause exponential mixing. For synergistic interactions where only higher order, rare combinations of variables are predictive, we show that early checkpoints correspond to low degree projections that reveal modes of $\pi^*(x)$, converting random-walk behavior with exponential mixing times to near-linear. Our method **works as-is**, requiring no additional compute or training changes. We demonstrate strong empirical gains on synthetic tasks, sampling from discrete energy-based models, and challenging real-world DNA and materials design tasks. *To the best of our knowledge, this is the first work to leverage a neural network's training trajectory to improve sampling.*

## 2   RELATED WORK

**Smoothing for sampling** There is a long line of work that smooths the target to accelerate sampling. Prominent examples include reward-conditioned diffusion and discrete walk–jump schemes that walk on a smoothed manifold and jump back to the discrete space (Yuan et al., 2023; Frey et al., 2024). Kirjner et al. (2024) train graph-smoothed protein fitness models ( 250K-node sequence graphs), and Zhu et al. (2025) show this smoothing induces a spectral bias that disproportionately damps high-degree Boolean monomials. Our approach instead leverages the *natural* smoothing in a network's training trajectory; we focus on the no-extra-compute regime and therefore do not benchmark against explicit smoothing methods.

**Interpretability** Many approaches probe a trained predictor via sampling/optimization. Minimal counterfactual edits are typically posed as constrained optimization on the fixed model,with sparsity or proximity regularization (Verma et al., 2024). A complementary line samples the predictor to quantify rare events—e.g., using MCMC to estimate the mass of inputs that violate a property or elicit outlier predictions (Webb et al., 2019). However, these approaches inherit the exponential barriers from rugged, high-frequency landscapes and random-walk exploration for synergistic interactions - issues our method mitigates when substituted for the sampling/optimization.

**Test-Time MCMC Sampling** Existing test-time, plug-and-play MCMC methods use gradient-guided methods for search and temperature annealing to overcome barriers. However, such temperature annealing methods (e.g. parallel tempering, annealed importance sampling, etc.) cannot bypass the random walk exploration that occurs from rare synergies (Hénin et al., 2022). Furthermore, for functions with high barriers, tempering relaxes barriers but offers little directional guidance - leading back to the random walk exploration. Thus, mixing time in the above settings remains exponential. Recent methods such as Diffusive Gibbs Sampling introduce an auxiliary noisy variable and alternate Gaussian noising with gradient-based denoising steps in a Gibbs scheme to improve mixing

on multi-modal targets (Chen et al., 2024). Similarly, Iterative Reasoning through Energy Diffusion (IRED) learns annealed energy landscapes but still depends on local energy gradients at test time (Du et al., 2024). However, all test-time gradient based methods are limited by the informativeness of local energy gradients.

**Discrete sampling** Grathwohl et al. (2021) introduced Gibbs-with-Gradients (GWG), using model gradients to choose which coordinate to flip rather than sampling indices uniformly. Since then, a flurry of work has pushed discrete MCMC forward along complementary axes: locally balanced/informed proposals that improve Metropolis–Hastings tuning (Zanella, 2017; Sun et al., 2022); non-local or parallel gradient moves via discrete Langevin and related formulations (Zhang et al., 2022; Sun et al., 2023); automatic cyclical scheduling of gradient-based updates for better mixing and reduced tuning (Pynadath et al., 2024); MALA-inspired discrete kernels with auxiliary-variable preconditioning (Rhodes & Gutmann, 2022); and reheated gradient-based samplers tailored to difficult combinatorial objectives (Li & Zhang, 2025). Our method is compatible with all of these gradient-based discrete kernels and could be combined with their proposal mechanisms or schedules; for simplicity, we use GWG throughout our discrete experiments.

**Coarse-to-Fine Learning** A growing body of theory suggests that gradient-based training and sampling in high-dimensional models proceeds in a coarse-to-fine manner. In diffusion models, linear and Gaussian analyses show that high-variance or low-frequency modes of the data covariance are learned and expressed in samples much earlier than low-variance, fine-detail modes, leading to an ordered emergence of global structure before local detail (Wang, 2025; Wang & Vastola, 2024). Related analyses of SGD on neural networks reveal multi-phase, saddle-to-saddle dynamics in which low-complexity or small-support features are acquired first, progressively enabling the learning of higher-order interactions (Abbe et al., 2023). Similar spectral decompositions of the NTK further indicate that only a few dominant eigendirections are amplified early in training, biasing learning toward coarse structure before finer modes are fit (Murray et al., 2022).

## 3 METHODS

**Test-time setting.** We work in a plug-and-play regime with a trained predictor $f^*: \mathcal{X} \to \mathbb{R}$ (MSE-trained on $\{(x_i, y_i)\}$), and we sample from its induced density $\pi^*(x) \propto \exp\{f^*(x)\}$, optionally under hard constraints. We do **not** compare against setups that modify training or fit auxiliary generative/score models (e.g., diffusion); our contribution is entirely in the test-time sampling procedure.

**Trajectory annealing.** Rather than run MCMC only on $\pi^*$, we traverse checkpoints along the training trajectory $\{f_t\}_{t=0}^T$ with $f_T \equiv f^*$, defining intermediate targets $\pi_t(x) \propto \exp\{f_t(x)\}$. Starting from $t = 0$, we apply a short Markov kernel for $N_t$ steps targeting $\pi_t$, carry the resulting state forward as the initializer for $\pi_{t+1}$, and continue this coarse-to-fine progression until $t = T$. For kernels, we use GWG+MH (Gibbs w/ Gradients + Metropolis Hastings) for discrete $x$ and MALA (Metropolis-Adjusted Langevin Algorithm) for continuous $x$.

Neural networks learn *coarse→fine*: low-frequency structure emerges early, high-frequency later. In discrete models trained with SGD, gradients align more with lower-degree monomials, so those coefficients converge first. In continuous models in the NTK regime, kernel eigenvalues decay with spherical-harmonic degree; the predictor is $f^*$ convolved with the kernel, giving strong early smoothing that relaxes over time.

### 3.1 BOOLEAN VARIABLES

Abbe et al. (2023) show SGD learns Boolean functions hierarchically: low-degree monomials are learned first as fewer variables leads to greater gradient alignment. We leverage this to turn *exponential* sampling into *polynomial* time. We study two hard classes: (i) $f^*$ that are hard because of high-frequency, high-magnitude variation—here, early checkpoints haven't learned the high-degree spikes yet, so the landscape is smooth and mixes quickly; and (ii) $f^*$ that are hard because there's no variation (the needle gadget). For needles, local sampling is a random walk - exponential in the needle dimension $d$. However, mixing on $f^*$ projected to monomials of degree $\leq 2$ mixes in $O(d \log d)$, with mixing worsening as the largest degree increases. The low-degree projection of $f^*$ also acts as an associative memory that can store many needles.

### 3.1.1 BACKGROUND: LOWER DEGREE MONOMIALS ARE LEARNED FIRST

Abbe et al. (2023) formalize a hierarchy in how SGD fits sparse Boolean targets. Writing the target as a sum of Boolean monomials, they define the *leap* as the smallest $k$ for which one can order the nonzero monomials so that, when adding the next monomial in that order, the union of involved variables introduces at most $k$ new variables. A pure "staircase" target—each term extending the previous by one fresh variable (e.g. $x_1 + x_1 x_2 + x_1 x_2 x_3$) -has leap 1.

This notion predicts hierarchical learning under SGD. For staircase-like functions, *lower-degree monomials are learned first and higher-degree monomials later*: initial gradients correlate more strongly with terms that require fewer new variables, so SGD first aligns a small set of coordinates; that alignment then amplifies gradients toward the next monomial, and so on. The trajectory passes through saddle-to-saddle plateaus; a phase that requires acquiring $L$ new variables at once takes $\tilde{\Theta}(d^{\max(L-1, 1)})$ steps, so the training time is dominated by the largest leap (i.e., the hardest stage).

They prove this in a *restricted* setting—two-layer fully connected networks with smooth activations, trained on i.i.d. data using a modified SGD (layer-wise updates plus a projection step)—and are complemented by empirical evidence: loss curves for deeper networks on hypercube data exhibit clear plateaus and drops consistent with learning across successive leaps (Abbe et al., 2023).

We provide additional empirical evidence for the hierarchical-learning picture across fully connected and convolutional networks, spanning a variety of activations, widths, and depths in Appendix C. Two regularities emerge: (i) *lower-degree* Fourier–Walsh components finish aligning with the target function earlier than *higher*-degree components, and (ii) the degree-wise mass grows only after all its monomials are fully aligned. See Fig. 1 for an example. A caveat is transformers, where we observe experimentally they satisfy (i) but not (ii). A degree-2 monomial could become aligned and grow in mass before all degree-1 monomials were aligned.

**Core assumption (degree-wise checkpoints).** We assume the setting in Abbe et al. (2023) holds for the larger networks we consider. Specifically, along the training trajectory $\{f_t\}_{t=0}^T$ with $f_T \equiv f^*$, there exist increasing checkpoints $\tau_0 < \tau_1 < \cdots < \tau_K \leq T$ such that at $\tau_k$ the model has effectively learned all interactions up to degree $k$, while higher-degree components are still negligible. Equivalently, we may treat $f_{\tau_k}$ as the degree-$k$ projection of the final model:

$$f_{\tau_k} \approx f_{\leq k}, \qquad f_{\leq k}(x) := \sum_{\substack{S \subseteq [d] \\ |S| \leq k}} \hat{f}^*(S) \prod_{i \in S} x_i.$$

Between these checkpoints, higher-degree terms may be partially learned; we assume only the existence and monotone ordering of $\{\tau_k\}$.

### 3.1.2 HIGH-MAGNITUDE, HIGH-FREQUENCY VARIATION

Early checkpoints in the training trajectory suppress high-degree terms, smoothing the landscape and making it easy to mix. We exploit this to handle targets with large high-degree components.

As a running example, consider $x \in \{\pm 1\}^d$ with

$$\pi_\gamma(x) \propto \exp\Big( \sum_{i=1}^d x_i + \gamma \prod_{i=1}^d x_i \Big),$$

where the linear term favors many $+1$ entries and the parity term $\prod_i x_i$ creates a high barrier when $|\gamma|$ is large.

At low temperature, vanilla Gibbs on the full objective mixes in **exponential** time $\tilde{\Theta}(\exp\{c|\gamma|\})$: once a random start flips to satisfy the parity term, any move that increases the number of $+1$ bits must cross a $|\gamma|$-sized barrier, so the chain gets stuck near suboptimal states.

Our trajectory sampler avoids this. We first run a short chain at checkpoint $\tau_1$; this mixes in $O(d \log d)$ under Gibbs (see App. A) and quickly reaches states with many $+1$ entries. We then continue the chain at the final checkpoint to adjust the parity. Thus, we are able to **hit the global maxima in near-linear time - sidestepping the exponential barrier.**

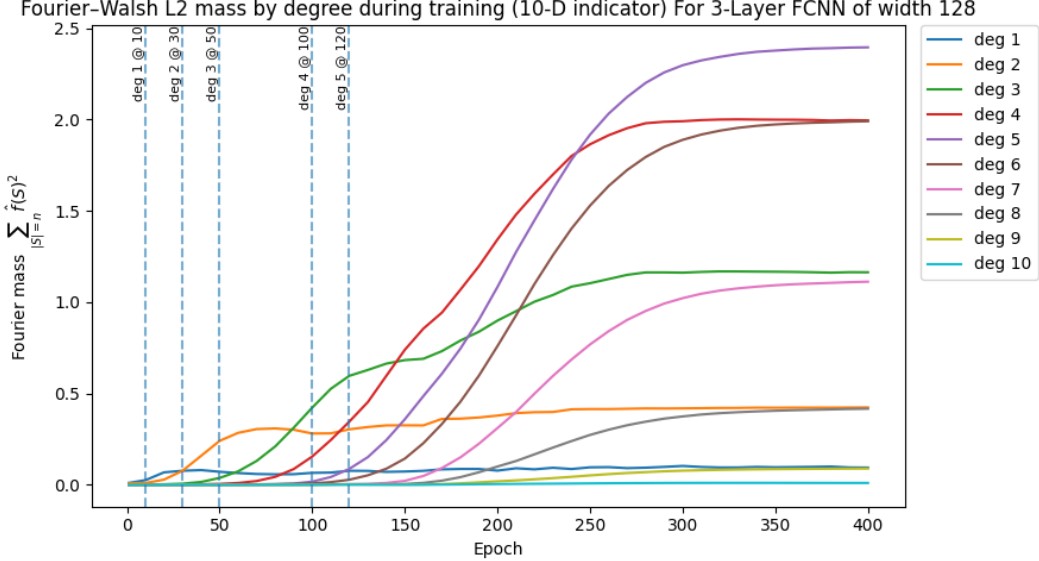

Figure 1: **Fourier–Walsh $L^2$ mass by degree during training on a $d{=}10$ indicator.** The target is $f(x) = \mathbb{1}\{x = z^\star\}$ with $x_i, z_i^\star \in \{\pm 1\}$. In $\{\pm 1\}$-coordinates this expands as $f(x) = 2^{-d} \sum_{S \subseteq [d]} \prod_{i \in S}(z_i^\star x_i)$, so the degree-$k$ component consists of all $k$-way monomials $\prod_{i \in S}(z_i^\star x_i)$ aligned with the pattern $z^\star$. Curves show $\sum_{|S|=k} \widehat{f_t}(S)^2$ over epochs for a 3-layer FCNN (width 128). Vertical dashed lines mark the *alignment epoch* for each degree $k$: the first epoch at which the sign of every degree-$k$ coefficient agrees with the sign implied by its aligned monomial (i.e., all signs point toward $z^\star$). We observe a consistent ordering: each degree first becomes aligned and then its Fourier mass rises, with *lower degrees* aligning and growing *earlier* than higher degrees. Although some *final* masses are larger at higher degrees (there are $\binom{d}{k}$ degree-$k$ monomials), this effect does not change the early-epoch ordering—low-degree components begin to align and increase first.

### 3.1.3 Synergistic interactions

**Needle-like synergistic interactions.** In our setting, variables interact *synergistically*: payoff arises only when a specific joint configuration is met (a "needle gadget"). Let $z^\star \in \{\pm 1\}^d$ denote that pattern and define the indicator

$$f^*(x) = \mathbf{1}\{x = z^\star\}, \qquad x \in \{\pm 1\}^d.$$

Over the Boolean (Walsh) basis, $f$ decomposes into all $2^d$ monomials aligned with $z^\star$:

$$f^*(x) = 2^{-d} \prod_{i=1}^{d} \left(1 + z_i^\star x_i\right) = 2^{-d} \sum_{S \subseteq [d]} \prod_{i \in S} \left(z_i^\star x_i\right)$$

so the degree–$k$ component is the sum of all $k$–way monomials $\prod_{i \in S}(z_i^\star x_i)$ with $|S| = k$.

Because the density is flat off of a tiny set $C$ (e.g., a single configuration), a local Markov chain behaves like a simple random walk on the $2^d$–vertex hypercube until it enters the 1–Hamming neighborhood of $C$. This is **exponential** in $d$.

**Intermediate checkpoint (degree-2 aligned)** Now suppose we are at checkpoint $\tau_2$. Let aligned spins $y_i := x_i z_i^\star$. The degree-$\leq 2$ surrogate can be written as

$$f_{\tau_2}(y) \approx f_{\leq 2}(y) = 2^{-d} \left( \sum_{i=1}^{d} y_i + \sum_{1 \leq i < j \leq d} y_i y_j \right), \tag{1}$$

which is the Curie–Weiss Hamiltonian with a positive external field up to scaling. Existing results show that in the low-temperature regime, we can hit $z^\star$ with high probability after $O(d \log d)$ steps

with a constant number of parallel chains. See App. B for details. **By exploiting the checkpoint $\tau_2$, we have converted the exponential random walk into near-linear mixing.**

**Multiple Needles** Even with multiple needles, we show theoretically (via connections to binary Hopfield models) and experimentally that low-degree monomials are sufficient to store and retrieve needles. Further details are in App. D.

**All checkpoints help** Sampling against $f_{\leq k}$ shows a monotone pattern: as $k$ grows during training, the landscape sharpens and the *needle hitting time increases*. Empirical evidence is in App. E. Sampling is fastest with $k = 2$ but sampling from any intermediate checkpoint with $k < d$ helps.

## 3.2 CONTINUOUS VARIABLES

Let $f^\star : S^{d-1} \to \mathbb{R}$ and let $t$ denote the time parameter. Gaussian (diffusion) smoothing on the sphere acts degree-wise on spherical harmonics: at time $t$, the degree-$k$ *coefficient* of $f_t$ equals the degree-$k$ coefficient of $f^\star$ multiplied by $M_k(t) = \exp\{-t\,k(k + d - 2)\}$ (larger $t \Rightarrow$ more smoothing; higher $k \Rightarrow$ stronger damping). NTK training (idealized FCNN: infinite width, zero init, uniform data) also acts degree-wise on spherical harmonics: the scaling $M_k(t)$ decays with degree $k$ as determined by the activation, $\Theta(k^{-d})$ for ReLU and $\Theta\big(k^{-d}e^{-\sqrt{k}}\big)$ for Tanh (Murray et al., 2022). **Takeaway.** The NTK training trajectory $\{f_t\}$ already provides a continuum of smoothed versions of $f^\star$—the same coarse-to-fine effect as heat-kernel smoothing, unlike diffusion which learns these smoothed functions explicitly. See App. F for more details.

## 4 RESULTS

We evaluate our method under matched compute on sampling from discrete functions - synthetic boolean functions (ruggedness, synergy), binary MNIST-EBM, DNA design task (including constrained sampling), and continuous functions - Ackley 10-D, and a superconductor design task.

### 4.1 DISCRETE SAMPLING EXPERIMENTS

Experimental details are in App. I.

### 4.1.1 SYNTHETIC BOOLEAN EXPERIMENTS

We conceptually show how our method can turn exponential mixing into near-linear on synthetic targets. It succeeds by leveraging *(1) fast mixing on low-degree surrogates* and *(2) knowledge of the function's support gleaned from those surrogates*.

We evaluate four functions: (i) targets dominated by high-degree components, (ii) indicator functions of increasing size, (iii) indicators with adversarial non-convex linear terms, and (iv) multiple indicators (synergistic interactions). To stress (2), every task includes 500 spurious variables.

To make the contrast stark, we run GWG on the final checkpoint for up to **2000** steps, while our method uses only **40** steps total.We report the percentage of runs that reach the global optimum. **Even with 50× more steps (2000 vs 40), GWG's hit rates remain very low (e.g. $\leq 3\%$), whereas our method is near-perfect on most tasks.**

This is because GWG is a random walk that succeeds mainly when it starts near the target, and that probability collapses exponentially with growing indicator size, more distinct synergies to satisfy, and many spurious variables. In contrast, our method's (1) fast mixing and (2) support knowledge do not degrade in these regimes.

Table 1: Sampling with 500 spurious variables on an 8-variable polynomial target dominated by high-frequency variation $f(x) = 0.1 \sum_{i=0}^{7} x_i + 0.2 \sum_{i=0}^{3} x_{2i} x_{2i+1} + 0.4 \sum_{i \in \{0,4\}} \prod_{k=0}^{3} x_{i+k} + 0.8 \sum_{i \in \{0,2\}} \prod_{k=0}^{5} x_{i+k} + 3.0 \sum_{i \in \{0\}} \prod_{k=0}^{7} x_{i+k}$. Success = hits all +1 global max. on 8 variables. *Result*: Ours outperforms both baselines (0.52 vs. 0.04) with a fraction of steps.

| Ours: Success (95% CI) | GWG: Success (95% CI) | GWG + Temp Anneal: Success (95% CI) |
|---|---|---|
| 0.5200 (0.45–0.59) | 0.01 (0.00–0.02) | 0.04 (0.01–0.06) |

**High-magnitude, high-frequency variation**  We construct a synthetic function whose coefficients increase with degree. With only 40 steps (vs 2000 for the baseline), we hit the global maxima 52%, compared to 4% for temperature-annealed GWG on the final checkpoint. See Table 1. This is because the early checkpoint allows for fast mixing without the high-frequency variation.

Table 2: Sampling with 500 spurious variables and $d$ indicator variables. **Ours** = 20 GWG steps at epoch 25 plus 20 at final. **Baseline** = GWG on the final checkpoint for 2000 steps. All runs use GWG with $\beta = 10$. Reported are success fractions with 2SD CIs; Success = hits indicator. Baseline also reports median steps to first hit (conditional on success; CI lower bounds clipped at 0).

| | Success probability (95% CI) | | GWG: median steps to first hit (given success) [95% CI] |
|---|---|---|---|
| $d$ | **Ours** | **GWG** | **Med. steps** |
| 3 | 0.98 (0.96–1.00) | 0.47 (0.40–0.54) | 1 [1–1] |
| 5 | 1.00 (1.00–1.00) | 0.21 (0.15–0.27) | 1 [1–1] |
| 8 | 1.00 (1.00–1.00) | 0.17 (0.11–0.22) | 4 [0–22.9] |
| 10 | 0.99 (0.98–1.00) | 0.12 (0.07–0.16) | 2 [0–56.8] |

**Indicator function**  For an indicator function with 500 spurious variables, a random walk takes, on average, $500 \times 2^d$ **steps**. GWG's conditional median steps (given a hit) are 1–4 steps, indicating it mostly succeeds when the initialization is close to the target. Thus, its hits degrade as $d$ increases (12% for $d = 10$) whereas our method remains perfect (despite having only $\frac{1}{50}$ steps). See Table 2.

**Indicator function with adversarial non-convexity**  We add an adversarial degree-1 terms that are opposite to the indicator pattern. However, because the indicator dominates the stationary measure, the local field is still dominated by the indicator's low-degree expansion. Thus, the adversarial linear term has a minimal impact and our method has a perfect hit rate. See Table 3.

Table 3: Sampling in a non-convex binary landscape. The objective is an indicator on 10 designated variables that yields 10 only at the all-ones pattern (and 0 otherwise), plus a linear term $-0.1 \sum_{i=1}^{10} x_i$ on the same variables that pulls toward all $-1$s; 500 additional variables are spurious (no effect). Success = hitting the indicator. *Result:* Ours is near perfect as the non-convexity is not able to dampen the signal from the intermediate checkpoint

| Ours: Success (95% CI) | GWG: Success (95% CI) |
|---|---|
| 1.00 (1.00–1.00) | 0.080 (0.0416–0.1184) |

Table 4: Sampling with 500 spurious variables and 3 non-overlapping length-5 indicators. Success = hits all 3 indicators at once. *Result:* Ours is near-perfect, while the baseline is near chance. With only one length 5 indicator (see Table 3), success rate is 0.21, drops with more indicators.

| Ours: Success (95% CI) | GWG: Success (95% CI) |
|---|---|
| 1.00 (1.00–1.00) | 0.0250 (0.0029–0.0471) |

**Multiple indicator functions**  GWG for a length-5 indicator has a modest hit-rate of 21% (see Table 1), however, performance collapses to 3% when we have three length-5 indicators (on non-

overlapping subsets). However, our method remains perfect. This is because GWG depends on starting near a good basin (which becomes exponentially unlikely as synergies compound). However, our method leverages (i) fast mixing on lower-degree surrogates and (ii) knowledge of the support to consistently (100% vs 3%) find the solution in a fraction of steps. See Table 4.

### 4.1.2 SAMPLING FROM MNIST ENERGY-BASED MODEL

The phenomena emphasized above – high-frequency variation and many synergistic interactions – are common to real-world data. Thus, we test our method's ability to efficiently sample from energy based models (EBMs) trained on binary MNIST.

We train a binary EBM with GWG using the implementation from Grathwohl et al. (2021). At test time, we compare temperature-annealing from the final checkpoint (the baseline inference method used in Grathwohl et al. (2021), which we denote *Temp-GWG*) vs. annealing along the training trajectory (our method). For both methods, we report FID after either 1K or 10K GWG sampling steps. For our method, we evenly distribute the steps across 500 evenly spaced checkpoints.

Table 5: FID ($\downarrow$) on binary MNIST using LeNet features. Mean (std) over 10 bootstraps

| Method | 1K steps | 10K steps |
|---|---|---|
| Temp-GWG | 29.61 (0.239) | 21.12 (0.138) |
| Ours | **11.73** (0.284) | **5.49** (0.119) |
| Ground-truth | 0.01 (0.013) | – |

Controlling for the number of steps, we observe substantially better FIDs in Table 5. We provide ablations on the number of checkpoints in App. G, which show significant improvements over the baseline across a wide range of checkpoint counts. App. H contains random samples; ours are substantially sharper.

### 4.1.3 DNA DESIGN EXPERIMENT

Recently, CNNs have matched or exceeded transformers for both protein and DNA language modeling—consistent with the fact that local, repeated motif patterns align naturally with convolutional filters (Yang et al., 2024; Bo et al., 2025). Moreover, standard TF assays reach million-scale because short oligo are mass-processed, yielding abundant data with higher-order motif structure, so trained regression models are both accurate and difficult to sample from (Berger et al., 2006).

We train a CNN on a dataset that measures binding affinity to the transcription factor (TF) MAX (Badis et al., 2009). The dataset consists of 42K 60-mer sequences. For sampling, we sampled a random length-60 sequence and performed 60 steps of GWG (on the final checkpoint for the baseline, and across training checkpoints for our method).

Table 6 shows that **our median samples are over** $10^8$ **more performant** (fitness is on a log-scale), partially driven by the ease of sampling the TF's motif (79% for ours vs. 39% for baseline).

We also show **our method's robustness in the constrained sampling setting** where we want to sample within a Hamming ball of a fixed starting point. This mirrors common biological use cases such as identifying minimal gene perturbations, minimal amino-acid mutations, or minimal CRISPR/base-editor edits to boost activity.

Notably, this setting is not straightforward to handle with standard diffusion pipelines. Diffusion models support inpainting tasks conditioned on fixed portions of the final output, but enforcing an exact constraint such as remaining within a Hamming ball of a fixed sequence typically requires additional machinery (e.g., auxiliary guidance networks Shen et al. (2024) or SMC-style schemes Wu et al. (2023)) rather than a single pre-trained denoiser. By contrast, our method handles this naturally by enforcing the Hamming-ball constraint throughout the sampling process, simply by restricting the MCMC chain to the constraint set. This plug-and-play ability to impose new hard constraints at test time, without training any additional generative or reward model, is a practical advantage of our predictor-based approach in this application.

Table 7: Per-run metrics with 95% bootstrap percentile CIs (B=500) for constrained sampling. Sampling trajectory is restricted to always stay within a hamming distance of 7 from the starting point. Each run starts from a random length-60 DNA sequence, runs 60 mutation steps, and keeps the best; repeated 300 times. Diversity/novelty are recomputed per bootstrap. Percentiles are vs. the training set restricted to $y > 0$.

| Method | Fitness (median) | Pct. | Diversity | Novelty | Motif (%) |
|---|---|---|---|---|---|
| Ours | **7.42** [7.03, 7.63] | **99.45** [99.38, 99.47] | 45 [45, 45] | 33 [33, 33] | **63.0** [58.2, 68.0] |
| GWG | 2.09 [0.98, 3.06] | 91.89 [66.36, 96.65] | 45 [45, 45] | 33 [33, 34] | 31.3 [26.0, 37.3] |
| AISAutoTemp-GWG | $-0.32$ [$-0.79$, $-0.10$] | 0.00 [0.00, 0.00] | 45 [45, 45] | 33 [33, 34] | 3.0 [1.3, 5.0] |
| PT-GWG | 1.92 [1.51, 2.58] | 90.18 [83.25, 95.03] | 45 [45, 45] | 33 [33, 34] | 24.3 [19.3, 29.3] |

Table 8: Main and secondary metrics. Bracketed values are 95% CIs computed as Student-$t$ intervals across seeds on per-seed means. Secondary metrics for Superconductor are reported as median [IQR]. *Refs:* Ackley 0.0; Superconductor 185.0.

| Experiment | Method | Best | Mean [95% CI] | Novelty / Diversity [IQR] |
|---|---|---|---|---|
| Ackley (10D, $\downarrow$) | MCMC–Final | 8.5628 | 16.2164 [16.1559, 16.2770] | – / – |
| | SMC–Temp | 7.8595 | 16.3141 [16.2735, 16.3547] | – / – |
| | AISAutoTemp | 8.8096 | 16.3095 [16.2679, 16.3512] | – / – |
| | PT | 13.6225 | 19.8698 [19.4593, 20.2803] | – / – |
| | **SMC–Train** | **3.6942** | **13.3311 [12.0867, 14.5755]** | – / – |
| Superconductor ($\uparrow$) | MCMC–Final | 107.4 | 76.68 [76.42, 76.94] | 17.30 [4.98] / 11.76 [7.47] |
| | SMC–Temp | 80.7 | 21.98 [21.39, 22.56] | 34.39 [2.42] / 12.76 [2.69] |
| | AISAutoTemp | 107.4 | 24.42 [23.88, 24.97] | 35.66 [5.18] / 26.05 [5.28] |
| | PT | 107.4 | 25.97 [24.31, 27.62] | 35.29 [5.01] / 25.40 [4.91] |
| | **SMC–Train** | **318.4** | **155.2 [105.6, 204.8]** | 20.86 [3.64] / 16.60 [6.87] |

Table 7 illustrates the challenging task of finding a length-7 Hamming perturbation to a random length-60 DNA sequence to improve the sequence's fitness. Again, our method finds samples that are $10^5$ more performant, and are twice as likely (63% vs 31%) to contain the motif.

Table 6: Per-run metrics with 95% bootstrap percentile CIs (B=500). Each run starts from a random length-60 DNA sequence, runs 60 mutation steps, and keeps the best; repeated 300 times. Diversity/novelty are recomputed per bootstrap. Percentiles are vs. the training set restricted to $y > 0$.

| Method | Fitness (median) | Pct. | Diversity | Novelty | Motif (%) |
|---|---|---|---|---|---|
| Ours | **10.04** [9.74, 10.20] | **99.78** [99.77, 99.80] | 45 [45, 45] | 34 [33, 34] | **74.3** [69.7, 79.3] |
| GWG | 2.72 [1.60, 5.53] | 95.54 [85.24, 98.83] | 45 [45, 45] | 33 [33, 34] | 38.7 [33.3, 44.0] |
| AISAutoTemp-GWG | 0.56 [0.16, 0.85] | 42.86 [12.99, 59.72] | 45 [45, 45] | 33 [33, 33] | 4.0 [2.0, 6.3] |
| PT-GWG | $-0.79$ [$-1.17$, $-0.25$] | 0.00 [0.00, 0.00] | 45 [45, 45] | 33 [33, 34] | 13.0 [9.3, 16.7] |

## 4.2 CONTINUOUS SAMPLING EXPERIMENTS

**Setup.** We compare four samplers under matched compute: Sequential Monte Carlo with temperature annealing (SMC–Temp), Annealed Importance Sampling (AISAutoTemp), Parallel Tempering (PT), and our Sequential Monte Carlo with training-time checkpoints (SMC–Train). All share the same compute budget, with results averaged over 5 seeds. Sampling/budget details are in App. K; SMC and checkpointing specifics are in App. J.

**Ackley (10D).** Rugged continuous optimization on $[-10, 10]^{10}$; **SMC–Train** attains the top mean and best-of-set under matched compute with non-overlapping CIs (See Table 8).

**Superconductor.** A real-world benchmark in high-D materials design taken from the design benchmark in Trabucco et al. (2022). It has a rugged, non-convex, heavy-tailed landscape. Inputs

$\mathbf{x} \in \mathbb{R}^{87}$ encode element composition; the target $y$ is the critical temperature $T_c$ (K). **SMC–Train** achieves the highest mean and best-of-set $T_c$, exceeding the reference and all baselines (See Table 8).

## 5  DISCUSSION

**Discrete vs. Continuous** In continuous domains, early training in the NTK (linearized) regime induces frequency-selective smoothing; as training leaves the NTK regime, this smoothing fades. In discrete domains, the effect is stronger. Because boolean targets are learned low-to-high degree, there are less new variables in the high degree term, causing higher gradient alignment. This allows for faster learning - in fact, the number of steps is asymptotically optimal, matching Correlational Statistical Query (CSQ) lower bounds (Abbe et al., 2023). Other work suggests that SGD learns with an optimal number of steps in more general settings (Barak et al., 2022). *Thus, hierarchical learning arises naturally from SGD's inherent efficiency.*

**Architectures** *Continuous* For FCNNs, the NTK eigenfunctions are spherical harmonics, so spectral bias aligns directly with smoothness (low degree $\leftrightarrow$ larger eigenvalues); CNNs/ResNets inherit this (Geifman et al., 2022; Belfer et al., 2024). Transformers have different eigenfunctions, so this does not apply. (Hron et al., 2020). *Discrete* Hierarchical interaction learning under SGD is FCNN-specific and transfers to CNNs/ResNets (we report results for all three). Transformers, by contrast, learn interactions via the attention matrix, which is qualitatively different.

**Limitations** Our method does not apply to transformers. However, commonly deployed models in medicine, personalized recommendations, and decision support tend to be CNN/MLPs, as detailed in Section 1 – interpretability is crucial in these domains. In addition, for protein/DNA predictor models, CNNs outperform transformers in low-N fitness tasks (Dallago et al., 2021), and have recently exceeded transformers in pretrained protein/DNA language models (Yang et al., 2024; Bo et al., 2025). This is because local, repeated motif patterns align naturally with convolutional filters.

## 6  CONCLUSION

Sampling from a trained predictor $f^\star$ is important for interpretability and compute-efficient design. However, rugged and needle-gadget landscapes lead to exponential mixing times that standard, temperature-annealing based MCMC methods cannot overcome. We demonstrate our trajectory-annealed samplers bypass this barrier across (1) three common architectures (FCNNs, CNNs, and ResNets) ranging from 2-20 layers and across (2) diverse tasks such as synthetic stress-tests, real-world design tasks, and EBM sampling. We theoretically characterize our method's benefits, showing exponential $\rightarrow$ near-linear sampling improvements under idealized conditions. To our knowledge, we are the first to identify and exploit this training-trajectory lens for neural network sampling. We hope our analysis can spark further research on this topic, including extensions to transformers. Given the method's simplicity, we hope it can become a useful tool for efficiently probing predictors.

## 7  ETHICS STATEMENT

This work introduces a sampling procedure that reuses training checkpoints to improve efficiency when exploring a trained predictor's landscape. When paired with interpretability workflows, this can help surface spurious correlations, biases, and failure modes prior to deployment.

At the same time, any method that accelerates sampling or optimization over model scores has dual-use potential: it could make it easier to construct high-confidence but misleading inputs (adversarial examples or jailbreak prompts), search for harmful designs , or probe models in ways that risk model inversion or privacy leakage if training data contain sensitive information.

## 8  REPRODUCIBILITY STATEMENT

The code and data for all experiments are contained in the supplementary zip file. The only exception is the MNIST-EBM sampling experiments. Here, the GWG repo was used as-is, with minimal changes for our inference method. Those minimal changes are explained in I.3.2.

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

## A $O(d \log d)$ MIXING FROM $\tau_1$ CHECKPOINT

**Setting.** Let $\Omega = \{-1, 1\}^d$ and let

$$f(x) = \sum_{i=1}^{d} h_i \, x_i$$

be a sum of degree-1 monomials (linear function) on the Boolean hypercube. We want to sample from the Gibbs measure proportional to $\exp(f)$:

$$\pi(x) = \frac{1}{Z} \exp(f(x)) = \frac{1}{Z} \prod_{i=1}^{d} \exp(h_i x_i) = \prod_{i=1}^{d} \pi_i(x_i),$$

so $\pi$ is a *product* distribution with one-dimensional marginals $\pi_i(x_i) \propto \exp(h_i x_i)$. Consider random-scan single-site Gibbs: at each step pick $I_t \sim \mathrm{Unif}([d])$ and resample $X_t(I_t)$ from $\pi(\,\cdot\,|X_{t-1}(\neg I_t))$, which for this product target equals the marginal $\pi_{I_t}$.

**Claim.** For random-scan single-site Gibbs on a product target,

$$t_{\mathrm{mix}}(\varepsilon) \leq d \big( \log d + \log(1/\varepsilon) \big).$$

In particular, $t_{\mathrm{mix}}(1/4) \leq d(\log d + \log 4) = O(d \log d)$.

**Proof.** Let the *refresh time*

$$\tau_{\mathrm{ref}} = \min\{t : \text{each coordinate } i \in [d] \text{ has been selected at least once by time } t\}.$$

Because $\pi$ is a product, whenever coordinate $i$ is selected we resample it *fresh* from $\pi_i$, independently of everything else. Hence, at time $\tau_{\mathrm{ref}}$ we have resampled every coordinate from its marginal, so

$$X_{\tau_{\mathrm{ref}}} \sim \prod_{i=1}^{d} \pi_i = \pi.$$

This makes $\tau_{\mathrm{ref}}$ a strong stationary time, which implies

$$\big\| P^t(x, \cdot) - \pi \big\|_{\mathrm{TV}} \leq \Pr(\tau_{\mathrm{ref}} > t) \qquad \text{for all starting states } x \text{ and times } t \geq 0. \tag{2}$$

It remains to bound the tail of $\tau_{\mathrm{ref}}$. Each step picks a coordinate uniformly from $[d]$, so this is the coupon-collector process. For any fixed $i$,

$$\Pr\big( i \text{ was never chosen in } t \text{ steps} \big) = (1 - 1/d)^t \leq e^{-t/d}.$$

A union bound over the $d$ coordinates then gives

$$\Pr(\tau_{\mathrm{ref}} > t) = \Pr(\exists \, i \text{ unrefreshed}) \leq d \, e^{-t/d}. \tag{3}$$

Combining equation 2 and equation 3 and choosing $t$ so that $d e^{-t/d} \leq \varepsilon$ yields

$$t \geq d \big( \log d + \log(1/\varepsilon) \big),$$

which proves the claim. $\qquad\square$

## B $O(d \log d)$ MIXING FROM $\tau_2$ CHECKPOINT

At checkpoint $\tau_2$, let aligned spins $y_i := x_i z_i^\star$. The degree-$\leq 2$ surrogate can be written as

$$f_{\tau_2}(y) \approx f_{\leq 2}(y) = 2^{-d} \left( \sum_{i=1}^{d} y_i + \sum_{1 \leq i < j \leq d} y_i y_j \right), \tag{4}$$

which is the Curie–Weiss Hamiltonian with a positive external field up to scaling.

In the low-temperature regime, existing results show that *censored* Gibbs dynamics on just the degree-2 monomials (ignoring the degree 1 monomials) mixes in $O(d \log d)$ time (Ding et al., 2009). By censoring, we mean if a proposed update would make the alignment with the pattern negative, we reflect all of the update variables.

Without censoring and with a uniform start at low temperature and zero field, the chain falls into the $+y_i$ or $-y_i$ basin with probability $\approx \frac{1}{2}$ each; a positive field (as is the case in our setting) biases toward the $+y_i$ basin.

After scaling with $\beta = 2^d$, $f_{\leq 2}(z^\star)$ is at least $2d$ higher than all the other configurations. Thus, under the measure $\exp 2^d f_{\leq 2}(y)$, the target $z^\star$ is at least $e^{2d}$ more probable than all other configurations. Since there are $2^d$ configurations in total, the lower bound for $z^\star$'s likelihood is $\frac{e^{2d}}{2^d - 1 + e^{2d}} = 1 - e^{-\Theta(d)}$. Consequently, after $O(d \log d)$ steps the chain is at $z^\star$ with high probability (so a constant number of parallel chains suffices).

## C  EMPIRICAL EVIDENCE OF LOWER DEGREES ALIGNING/GROWING BEFORE HIGHER DEGREES

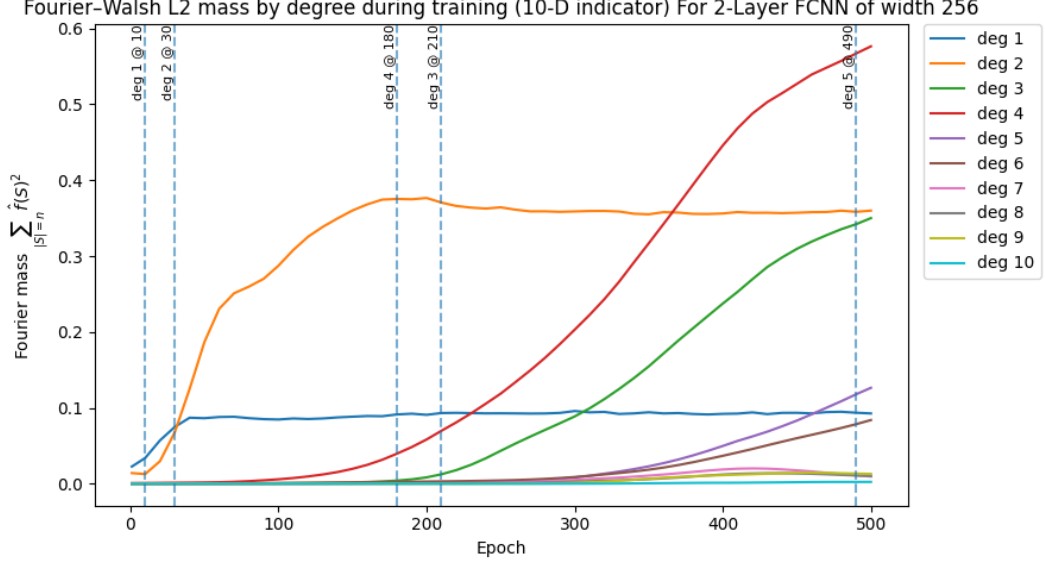

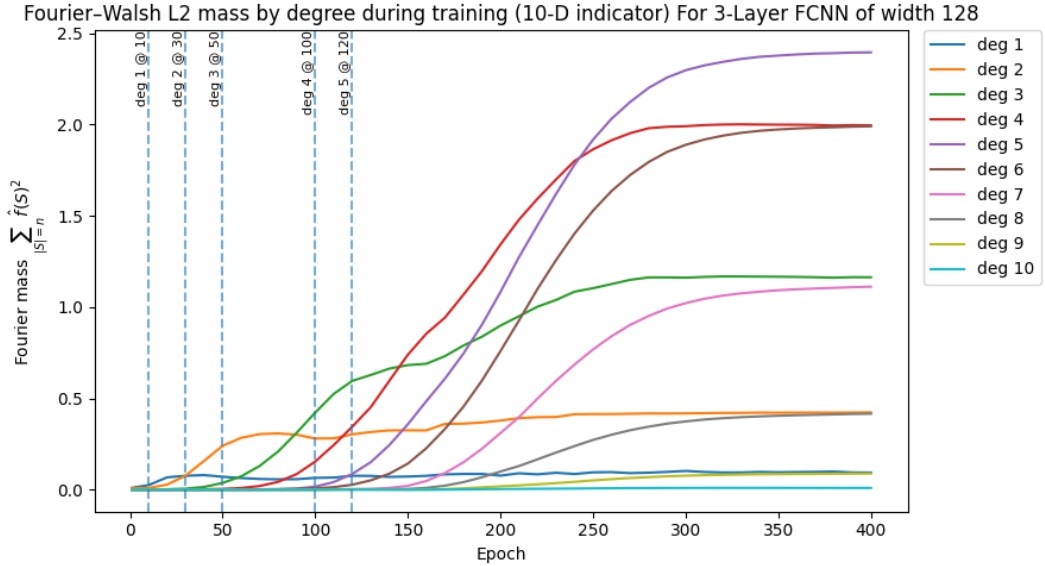

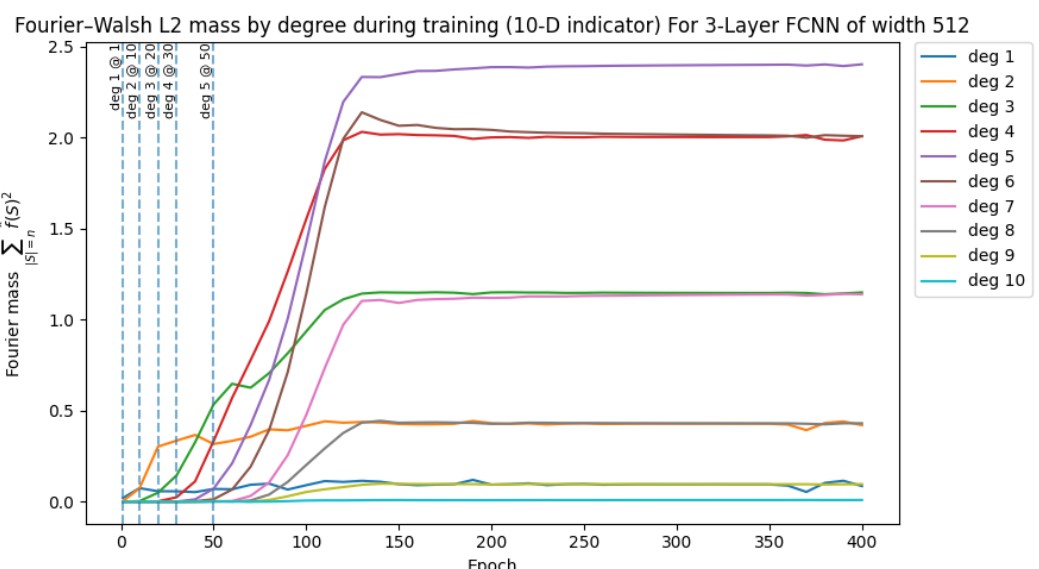

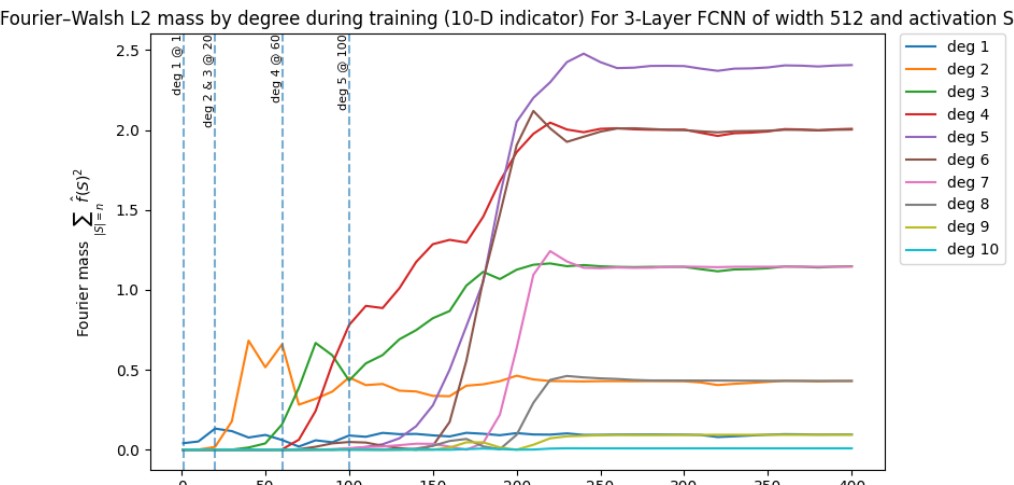

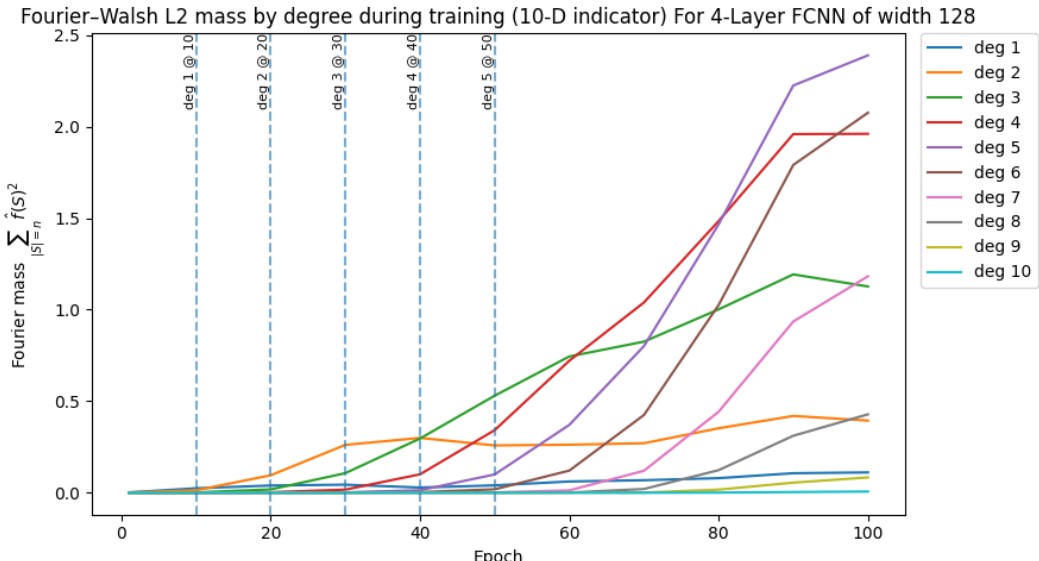

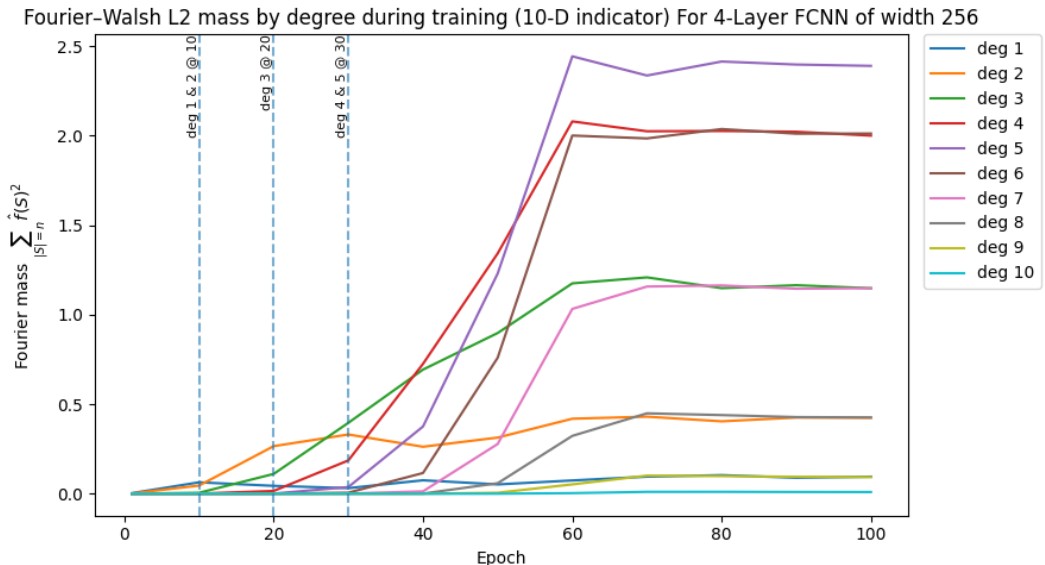

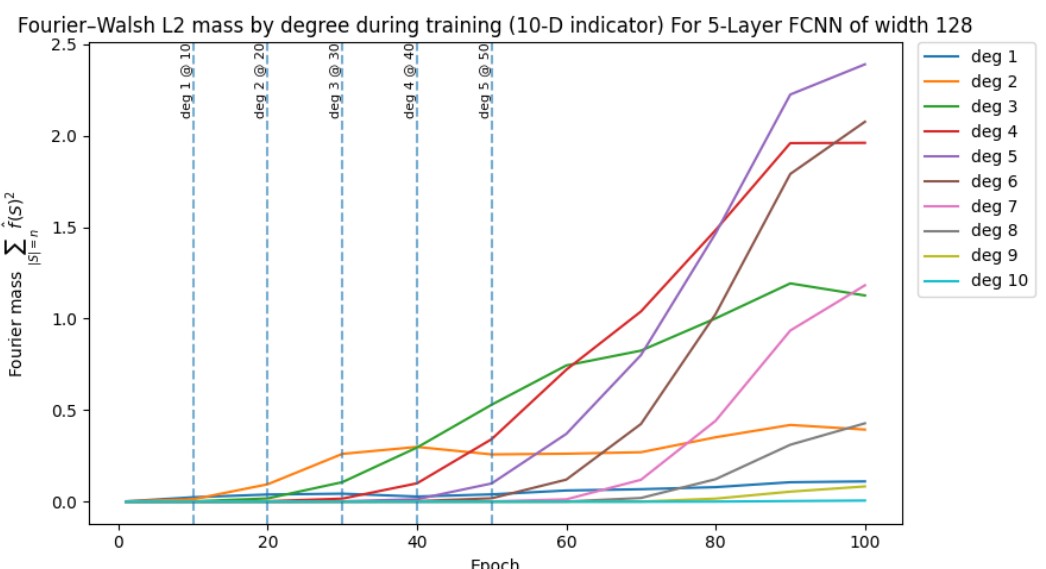

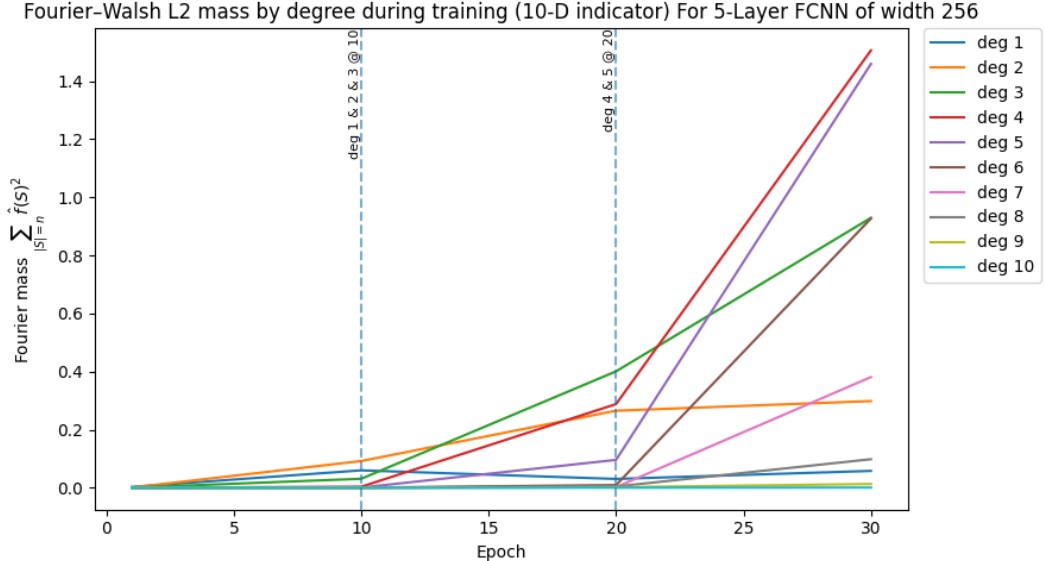

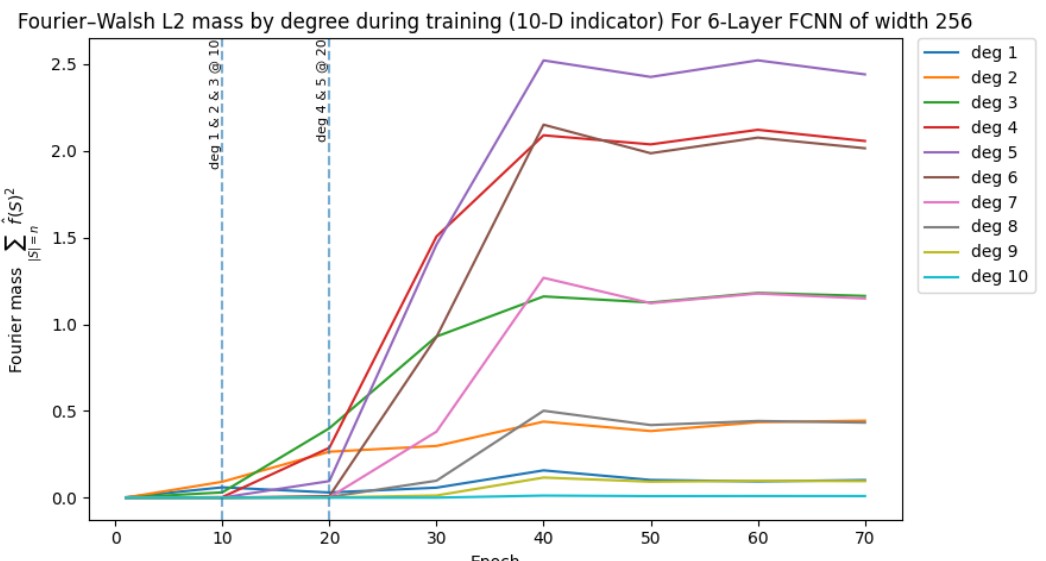

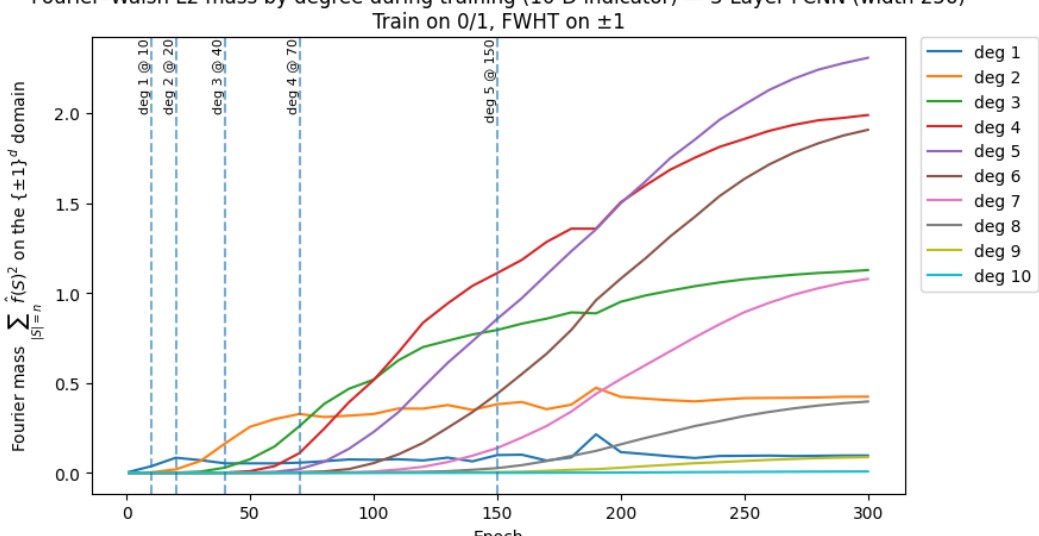

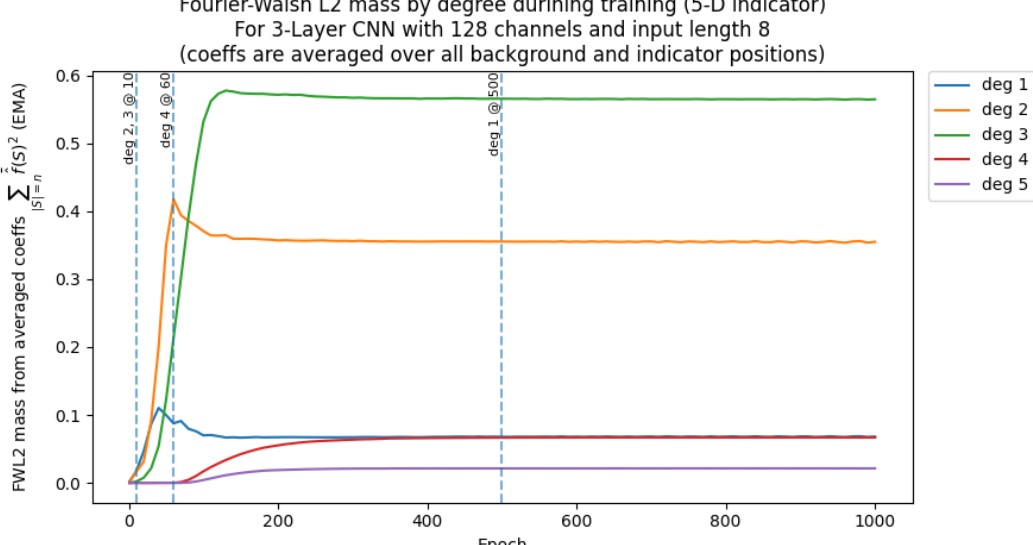

# D   MULTIPLE NEEDLES

With multiple needles, the degree-2 interaction is equivalent to a Hopfield model with weights set by a Hebbian learning rule; increasing the number of stored needles shrinks attraction basins, potentially slowing local MCMC (McEliece et al., 1987; Storkey & Valabrègue, 1999). For the standard pairwise Hopfield model with unbiased needles, the capacity scales linearly as $N_{\max} \approx \alpha_c d$ with $\alpha_c \simeq 0.138$ (Amit et al., 1985). For pure $p$-spin Hopfield models, the number of storable patterns scales as $N_{\max} \approx \alpha_p d^{p-1}$, with $\alpha_p$ a $p$-dependent constant (Bovier & Niederhauser, 2001).

Table 9 shows that across multiple needles, sampling along the training trajectory is still more sample-efficient than running Gibbs-with-gradient on the final trajectory.

Table 9: Needle gadget: hit rates across 5 runs for GWG vs. our method (1 needle hit). Task: 10-D indicator "needle" with 10 additional linear terms; model: 3-layer FCNN (width 128); sampling: 3 particles, 60 total steps per run across checkpoints {5, 25, 50, 75, 100, final} (10 steps per checkpoint); baseline: 60 steps of GWG on final checkpoint

| # Needles | GWG: runs with $\geq 1$ hit (out of 5) | Ours: runs with $\geq 1$ hit (out of 5) |
|---|---|---|
| 5 | 0/5 | 5/5 |
| 4 | 2/5 | 5/5 |
| 3 | 1/5 | 5/5 |
| 2 | 0/5 | 5/5 |
| 1 | 0/5 | 5/5 |

# E  EMPIRICAL EVIDENCE OF SAMPLING FROM INDICATOR FUNCTION PROJECTED TO TERMS OF ORDER $\leq P$

The below graphs contain the median number of steps needed to hit the target when sampling from an indicator function that is only non-zero for the target. However, its boolean expansion is projected down to terms of degree $\leq P$. We consistently observe that the median number of steps increases as $P$ increases. For each value of $P$, we select the best $\beta$ value across (1.0,0.5,0.3,0.2,0.1,0.07,0.05,0.03,0.02,0.01), conduct 300 trials, and cap the number of steps at $2^d$. Sampling is done via Gibbs (random-index heat bath).

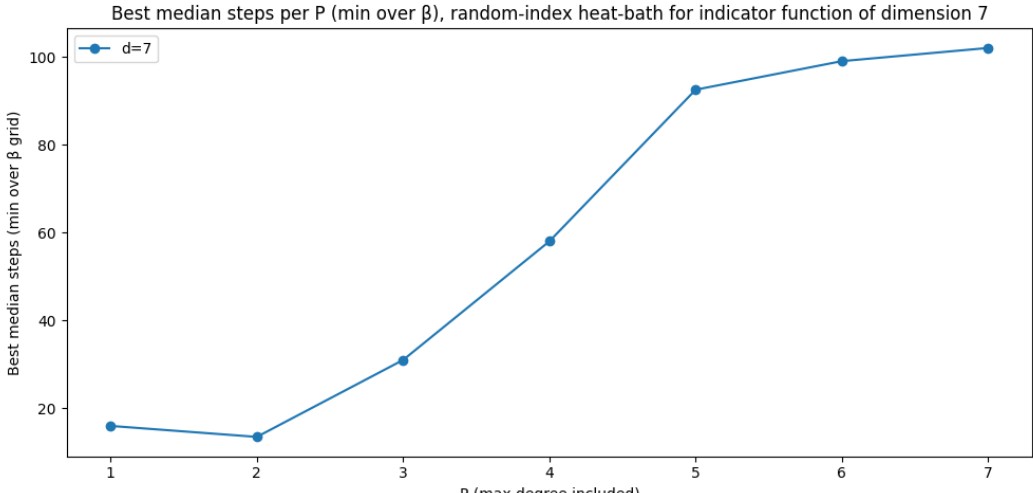

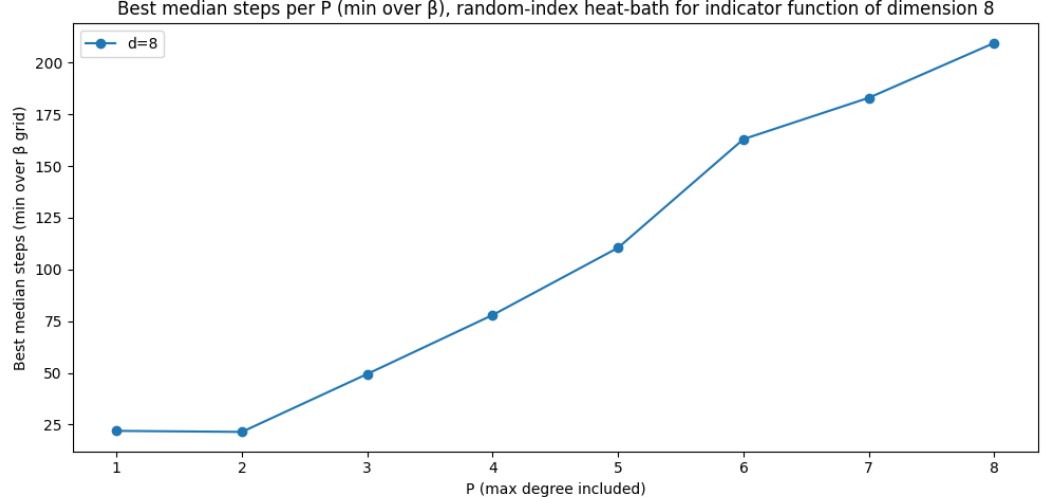

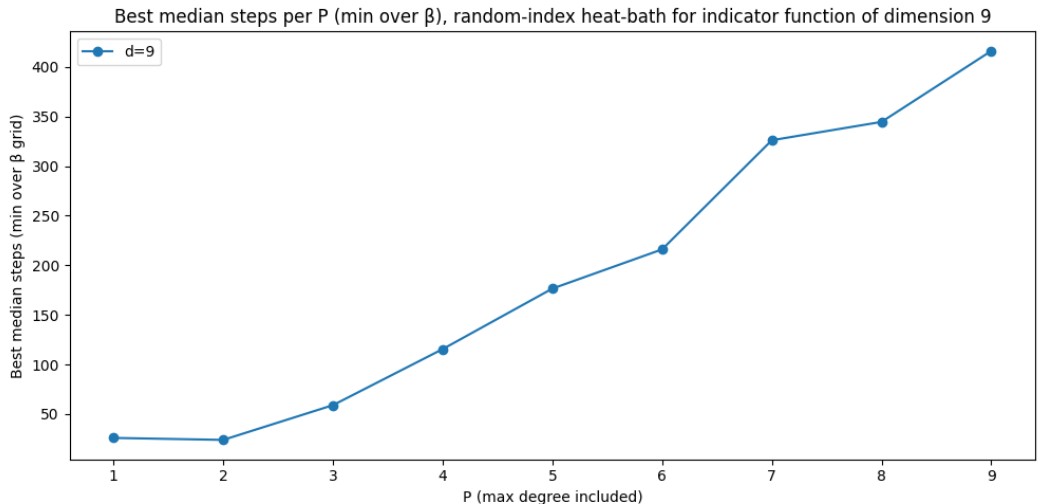

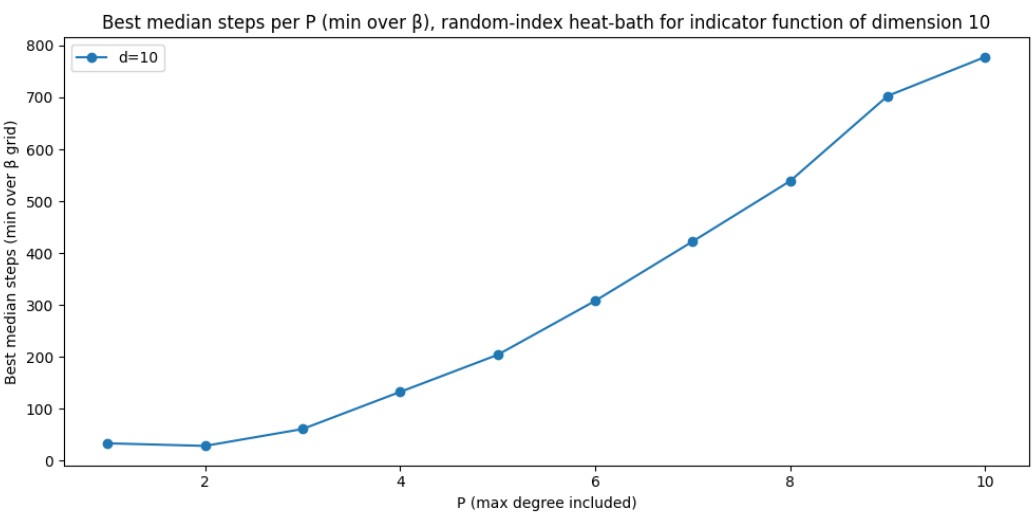

## F  NTK-ALIGNED TRAINING MAP

**Setup.**  Let the input domain be the unit sphere $\mathbb{S}^{d-1}$ with the *uniform* measure. Expand any square-integrable $f : \mathbb{S}^{d-1} \to \mathbb{R}$ in spherical harmonics $Y_{k,j}$ (degree $k \in \{0, 1, \dots\}$, multiplicity index $j$):

$$f(x) = \sum_{k=0}^{\infty} \sum_{j=1}^{N_k} a_{k,j} \, Y_{k,j}(x), \qquad a_{k,j} = \langle f, Y_{k,j} \rangle.$$

For isotropic (zonal) operators on the sphere, each degree-$k$ subspace is an eigenspace, so all coefficients $\{a_{k,j}\}_{j=1}^{N_k}$ evolve by the same scalar multiplier.

**Idealized NTK dynamics.**  Consider a fully connected network in the *linearized NTK* regime (infinite width), trained by kernel gradient flow with squared loss and learning rate $\eta$, from zero initialization, on data drawn uniformly from $\mathbb{S}^{d-1}$. The analytical NTK $K$ is a zonal kernel whose eigenfunctions are the spherical harmonics and whose degree-$k$ eigenvalue we denote by $\lambda_k > 0$. Along each degree-$k$ subspace the (prediction) coefficient obeys

$$\frac{d}{dt} \big( a_{k,j}(t) - a_{k,j}^{\star} \big) = -\eta \, \lambda_k \big( a_{k,j}(t) - a_{k,j}^{\star} \big),$$

so from $a_{k,j}(0) = 0$ we get

$$a_{k,j}(t) = \big( 1 - e^{-\eta \lambda_k t} \big) \, a_{k,j}^{\star}.$$

Equivalently, *at time $t$ the entire degree-$k$ block is scaled by*

$$M_k^{\mathrm{NTK}}(t) = 1 - e^{-\eta \, \lambda_k t} \in [0, 1].$$

This corresponds to Eq. (7) in Bowman (2023); see that reference for a fuller introduction.

Using $\frac{x}{1+x} \leq 1 - e^{-x} \leq x$ for $x \geq 0$ with $x = \eta t \, \lambda_k$, we obtain

$$\frac{\eta t \, \lambda_k}{1 + \eta t \, \lambda_k} \; \leq \; M_k^{\mathrm{NTK}}(t) \; \leq \; \eta t \, \lambda_k.$$

Hence for large $k$ (so $\lambda_k \to 0$),

$$M_k^{\mathrm{NTK}}(t) \sim \eta t \, \lambda_k,$$

i.e., $M_k^{\mathrm{NTK}}(t) \asymp \lambda_k$ up to constants depending on $\eta t$.

Activation choice controls the spectrum $\{\lambda_k\}$ and thus the decay of $M_k^{\mathrm{NTK}}(t)$ across degrees: for ReLU, $\lambda_k = \Theta(k^{-d})$ (polynomial "spectral bias"); for Tanh, $\lambda_k = \Theta\big(k^{-d} e^{-\sqrt{k}}\big)$ (super-polynomial). For fixed $t$ and large $k$, $M_k^{\mathrm{NTK}}(t) \approx \eta t \, \lambda_k$, so high degrees are damped more (Murray et al., 2022).

**Comparison: Gaussian (heat) smoothing.**  Heat-kernel smoothing on $\mathbb{S}^{d-1}$ multiplies the degree-$k$ block by

$$M_k^{\mathrm{heat}}(t) = \exp\{-t \, \mu_k\}, \qquad \mu_k = k(k + d - 2),$$

i.e., an exponential-in-$k^2$ decay (stronger high-frequency suppression). Note the *time* contrast: larger diffusion time $t$ means more smoothing, whereas larger NTK training time $t$ means $M_k^{\mathrm{NTK}}(t) \uparrow 1$ and less smoothing (the predictor approaches $f^{\star}$).

**Takeaway.**  Under the NTK idealization, the training trajectory $\{f_t\}$ is a family of degree-wise smoothed versions of $f^{\star}$, with the spherical harmonics as eigenfunctions and activation-controlled frequency decay. Diffusion performs a similar degree-wise smoothing but with heat-kernel multipliers.

## G  BINARY MNIST CHECKPOINT ABLATIONS

We report FID scores for varying numbers of checkpoints used in our method for a fixed budget of 1K total GWG steps. The model was trained for 50,000 epochs. For a given number of checkpoints,

we choose them to be evenly spaced along the training trajectory and allocate an equal number of sampling steps to each checkpoint. The Temp-GWG baseline samples only from the final checkpoint using temperature annealing.

Table 10: FID ($\downarrow$) on binary MNIST as a function of the number of checkpoints used by our method, with a fixed budget of 1K GWG sampling steps. Entries are mean (std) over 10 bootstrap resamples.

| # Checkpoints | Mean FID (std) |
|---|---|
| Temp-GWG (baseline) | 29.61 (0.239) |
| 5 | 16.10 (0.237) |
| 10 | 14.42 (0.322) |
| 25 | 12.56 (0.289) |
| 50 | 11.93 (0.355) |
| 100 | 12.08 (0.435) |
| 500 | 11.73 (0.284) |

As shown in Table 10, we see dramatic gains from using even 5 checkpoints, with additional checkpoints yielding diminishing marginal returns and performance saturating around 50–500 checkpoints. The key empirical observation is that the training trajectory tends to evolve from coarse to fine, so any set of evenly spaced epochs can leverage this structure to speed up sampling.

## H   SAMPLING FROM BINARY MNIST

Figure 17: First 49 random samples from standard sampling with 1K steps.

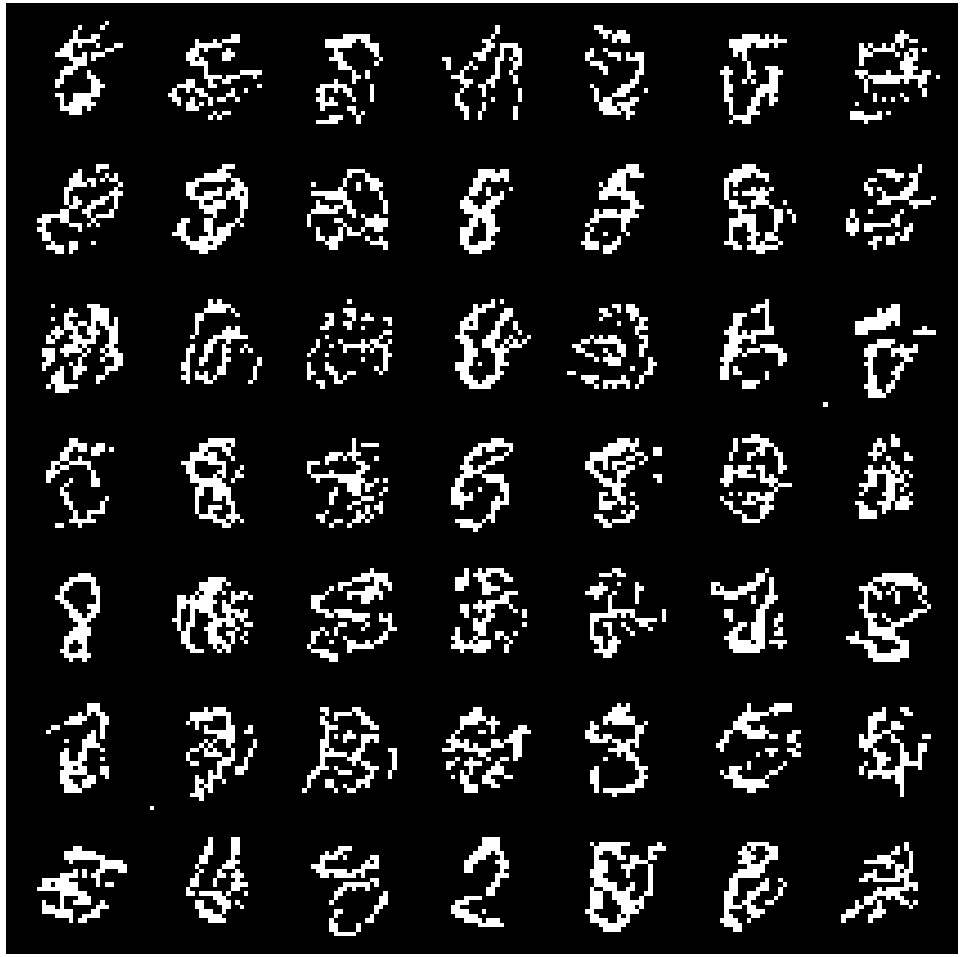

Figure 18: First 49 random samples from our method with 1K steps. These samples are substantially sharper than the above.

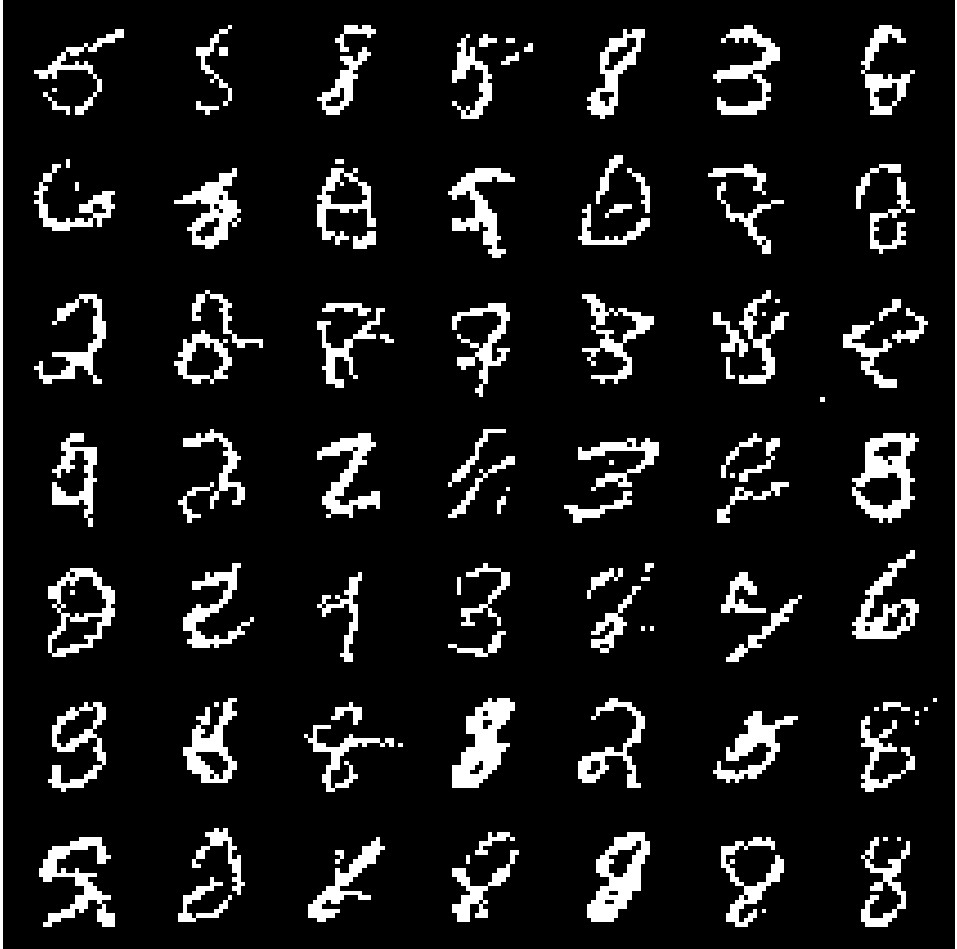

Figure 19: First 49 random samples from standard sampling with 10K steps.

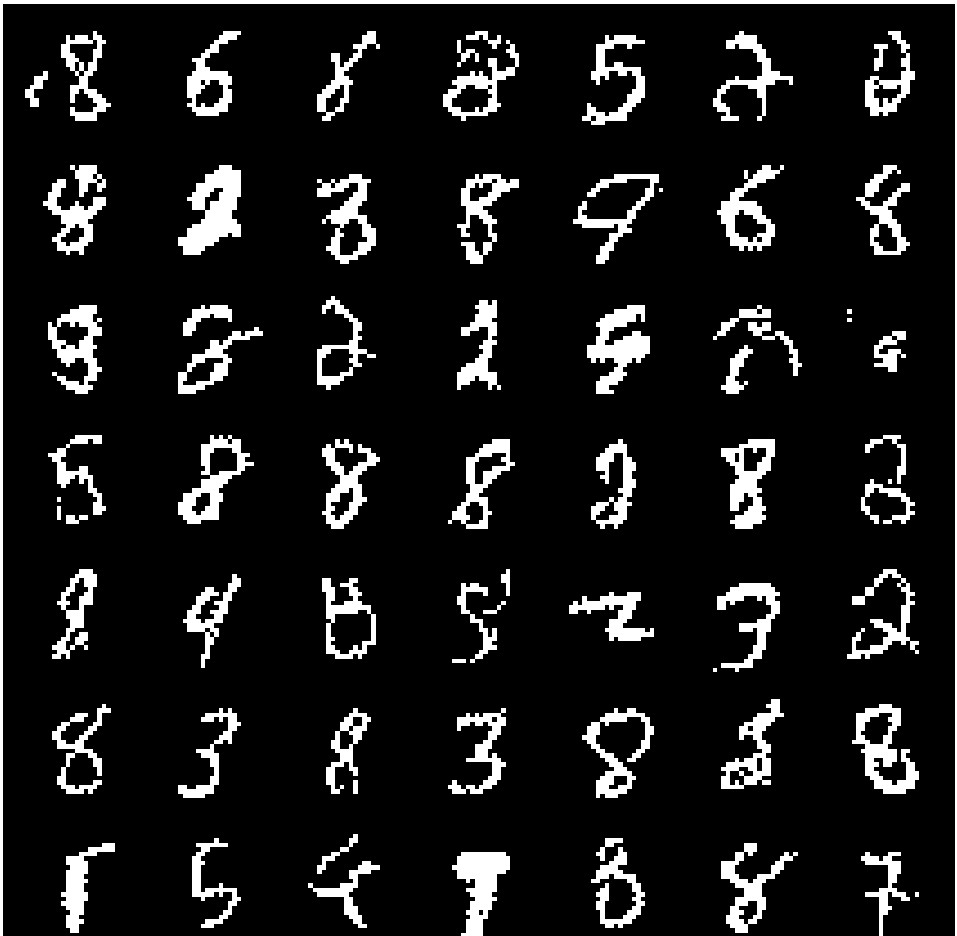

Figure 20: First 49 random samples from our method with 10K steps. Again, these samples are sharper.

# I    DISCRETE EXPERIMENTAL DETAILS AND COMPUTE BUDGETS

## I.1    CHECKPOINT AND TEMPERATURE SELECTION

For all experiments, we run an equal number of MCMC steps at each stage. We use checkpoints at epoch 25 and temperature = 10.0 for most experiments with a few exceptions detailed below.

For the DNA design experiment, we add an additional checkpoint at epoch 50 due to the additional complexity of the task.

For the discrete high-frequency experiment, we use temperature = 100.0 (instead of the 10.0 used in all other settings) because we are sampling from lower-magnitude component.

For the MNIST sampling experiment, we save the model every 100 epochs during training and use these as the checkpoints. We use the default temperature hyperparameters used in the repo; we make no changes.

## I.2    KERNEL

All experiments and baselines use the Gibbs-with-Gradients (GWG) kernel introduced in Grathwohl et al. (2021). This kernel uses gradient-informed proposals for categorical flips under a ±1 encoding, drawing moves from a softmax over approximate first-order energy changes. Each proposal is then corrected by a Metropolis–Hastings step that accounts for both the approximate energy difference

and the asymmetric forward/reverse proposal probabilities. The choice of temperature for the kernel is detailed in the subsection above.

For the synthetic experiments we use the *exact* GWG kernel where the $y$ values are the exact $y$ values, not those estimated via gradients. This is to remove the effect of gradient estimation noise. For the subsequent experiments, we use the standard GWG kernel where the $y$ values is approximated from the gradient.

For the MNIST sampling experiment, we use the kernel implementation from the repo (also GWG-MH); we make no changes.

For the constrained DNA sampling task, we modify the GWG-MH kernel so that it incorporates the Hamming distance constraint. If a sampled single-flip would exceed the Hamming cap, we pair it with a gradient-guided reversion on some already-flipped position so the net move stays on the Hamming boundary. Both legs are sampled from the same GWG softmax over their respective candidate sets.

### I.3 CHECKPOINT POLICY AND BASELINES

#### I.3.1 SYNTHETIC

We run 20 steps of our kernel at epoch 25, and then run another 20 steps at the final epoch. For the baseline, we run 2000 steps of the same kernel at only the final checkpoint. From random starting particles, we measure the fraction of particles that hit the global maxima of the function along their trajectory.

For the synthetic experiments, we only benchmark against GWG. This is because for the indicator function, other methods only manipulate the temperature of the final checkpoint. However, rescaling the temperature does not help - the landscape is still a random-walk.

For the High-magnitude, high-frequency variation experiment, we additionally benchmark against temperature annealing with a linear schedule. Here, the temperature was annealed from 0 to 100 (the temperature value used for this experiment) across 2000 steps, where one MCMC step was taken at each temperature value.

#### I.3.2 MNIST SAMPLING

We utilize the code, as-is, in GWG repo. This is trained for 50K epochs, and we save the checkpoint every 100 steps. For sampling with 1K steps, we run 2 steps of the kernel at each checkpoint. For sampling with 10K steps, we run 20 steps of the kernel at each checkpoint. We do not adjust any of the kernel hyperparameters for our task - we only change the checkpoints. For the baseline, we use the existing sampling code in the repo, which samples from the final checkpoint using linear temperature annealing and takes one kernel step for each temperature value. We control for the number of kernel steps across comparisons.

For the FID calculations, we use the repo https://github.com/abdulfatir/gan-metrics-pytorch/tree/master. We also use their MNIST LeNet model checkpoint for computing the metrics. We collect 8K random samples from the binary MNIST test set, our sampling method, and the existing sampling method. For the ground-truth FID value, we calculate FID between two random 8K subsets of the binary MNIST test set.

#### I.3.3 TF-DNA SAMPLING

We start from a random length-60 DNA sequence, and run 60 steps of the MCMC kernel. All methods only use 60 steps. Other than plain GWG, all of the methods use 3 stages with an equal number of steps for each stage.

For our method, we run 20 steps at epoch 25, 20 steps at epoch 50, and 20 steps at the final checkpoint.

For the GWG baseline, we run 60 steps at only the final checkpoint.

For the parallel tempering baseline, we run 3 replicas over a geometric $\beta$-ladder from $0.05 \rightarrow 10.0$, doing 20 local steps per replica with swaps every 5 steps (60 total local updates across replicas). For the replica starting point, the cold chain starts at $x_0$, while the mid and hot chains randomly reassign 25% and 50% of positions, respectively, to diversify exploration from the outset.

For annealed importance sampling with automated temperature adjustment, we run annealed importance sampling with an ESS-targeted adaptive temperature schedule ($\beta : 0 \rightarrow 10$) without resampling, preserving one-to-one lineages; each stage re-weights particles and picks the next $\beta$ to keep ESS near 0.6 N. After reweighting, it performs K=20 GWG rejuvenation sweeps at the new $\beta$; with 3 stages, this is 60 sweeps per particle.

For the constrained sampling task, we start from a random DNA sequence and want the best sample within a hamming distance of 7. We use the same setup as before except with a modified GWG-MH kernel detailed in Sec. I.2.

Diversity is calculated as the median of pairwise (Hamming) distances within each bootstrap resample. Novelty is calculated as median of per-seq min distance to training, within each bootstrap resample.

### I.4 ARCHITECTURE AND TRAINING DETAILS

For the synthetic experiment, we use a 3-layer FCNN with hidden dim 128. It is trained for 30000 epochs with a learning rate of 2e-3. Indicator functions have $y$ value of 10.

For the MNIST experiment, we use ResNet-EBM architecture specified in the GWG repo. The architecture is an EBM with a 3×3 stem conv, two strided residual downsampling blocks (each: Swish → 3×3 conv → 3×3 conv + a 1×1 projection shortcut), followed by six identity residual blocks (Swish → 3×3 → 3×3, no projection), then global spatial averaging and a single linear head to a scalar energy. In total, it has 19 conv layers and 1 fully connected layer. Training is done according to the repo's instructions.

For the TF-DNA experiment, we use the same architecture from de Almeida et al. (2022). Specifically, we apply a Conv1d(4→64, kernel=11), ReLU, then global max pooling over the sequence dimension for each filter, and then a linear head that outputs a scalar.

### I.5 DATASETS

For the synthetic experiments, we have 8-10 main variables that are part of the function and 500 spurious variables. We construct synthetic datasets where the spurious variables are randomly sampled.

We use the binary MNIST datasets provided in GWG repo.

For the TF-DNA experiment, the data consists of length-60 DNA sequences. The last 24 DNA letters are always the same; the first 36 are close to random. The TF (MAX, from the mouse species) binds strongly when the motif "CACGTG" is present. The binding strength increases depending on where the motif is present (upstream leads to stronger binding), the flanking sequences surrounding the motif, the number of times the motif is present, the GC % in the sequence, etc. The dataset is taken from Badis et al. (2009).

### I.6 CI DETAILS

All reported CIs are 2SD, unless specified otherwise.

For the synthetic experiments, we run the above test on 200 random particles, calculate the hit fraction (whether a particle reaches the global maxima along its trajectory),and report 2SD CIs from these results.

For the MNIST experiment, FID is calculated across sets of 8K samples. The standard deviation is calculated over 10 bootstraps.

For the DNA design experiment, we sample with 300 particles, and calculate 95% bootstrap percentiles (B=500) from the results.

## J   SAMPLING ALONG THE TRAINING PATH WITH SMC

Let $\pi_t(x) \propto \exp\big(f_t(x)\big)$ denote the (unnormalized) target associated with the checkpoint at time $t$ (e.g., from the model's energy or surrogate negative log-likelihood).

Rather than selecting a single smoothing level, we sample *along* the NTK training trajectory $t \in [0, T]$ using Sequential Monte Carlo (SMC):

1. Choose a schedule $0 = t_0 < t_1 < \cdots < t_L = T$ (e.g., geometric).
2. Initialize particles from an easy reference.
3. For $\ell = 1, \ldots, L$: compute incremental weights $w(x) \propto \pi_{t_\ell}(x)\big/\pi_{t_{\ell-1}}(x)$, resample, and apply a short MCMC move targeting $\pi_{t_\ell}$.
4. Output particles at $t = T$ (the desired final target).

This procedure exploits the frequency–selective filtering $M_k^{\mathrm{NTK}}(t)$ to traverse from a smooth–dominated intermediate distribution toward the final target while maintaining particle diversity.

## K   CONTINUOUS EXPERIMENTAL DETAILS AND COMPUTE BUDGETS

### K.1   SHARED SETTINGS (ALL EXPERIMENTS)

- **Parallel trajectories.** All methods use concurrent trajectories.
- **Compute parity.** Within each experiment, every trajectory performs the same total number of Metropolis–Adjusted Langevin Algorithm (MALA) steps across methods.
- **Kernels.** Ackley and Superconductor use MALA for all five methods.
- **Checkpoint policy (SMC–Train).** Train for 10,000 epochs; checkpoint every 10 epochs. Smooth the training-loss curve and keep the earliest prefix of checkpoints up to (but not beyond) the plateau; exclude later flat checkpoints (each kept checkpoint has strictly lower loss than the previous one).
- **SMC–Temp schedule.** Linear inverse-temperature ladder with the same number of distributions as SMC–Train for that task.
- **AIS schedule.** Annealed Importance Sampling (AIS) chooses temperatures adaptively each stage to maintain a target conditional effective sample size (cESS); rejuvenation uses the same MALA kernel as other methods.
- **PT schedule.** Parallel Tempering (PT) with a fixed temperature ladder across replicas; propose swaps between adjacent replicas every stage; within-replica moves are MALA with the same per-step budget.

### K.2   EXPERIMENT-SPECIFIC PARAMETERS

**Ackley (10D).**   **Proposal:** MALA with step size $10^{-2}$; adaptation target acceptance $0.57$ (adaptation off unless stated). *SMC–Temp:* resample when ESS $< 0.5N$. *SMC–Train:* resample when ESS $< 0.5N$. *AIS–Auto:* choose temperatures to hit cESS $= 0.5N$ per increment (bisection tol. $10^{-4}$, max 50 iters). *PT–MALA:* power temperature ladder (parameter $4.0$); per-replica MALA step scales as $\varepsilon/\beta^{1.0}$. *MCMC–Final:* plain MALA with the same step size; no burn-in, no thinning.

**Superconductor.**   Inputs $\mathbf{x} \in \mathbb{R}^{87}$. **Proposal:** MALA with base step size $\varepsilon = 0.05/\sqrt{d}$; adaptation target acceptance $0.57$ (off by default). **Stabilization in latent $z$:** per-dimension percentiles $[1\%, 99\%]$ and radial cap at $99.5\%$. *SMC–Temp / AIS–Auto:* resample / choose temperatures to maintain cESS $= 0.5N$; rejuvenation uses the same MALA step. *SMC–Train:* same cESS rule; default initialization from a Gaussian prior over $z$. *PT–MALA:* geometric temperature ladder up to $\beta_{\max} = 1.0$, swaps every stage; replicas chosen to evenly factor the parallel budget; report adjacent-swap rates and per-$\beta$ MALA acceptance. *MCMC–Final:* if adaptation is enabled: target acceptance $0.57$ with updates every 10 steps (clip $\varepsilon$ to $[10^{-4}, \, 0.5]$).

### K.3 COMPUTE BUDGETS

Table 11: Per-trajectory budgets. $L$ is the number of intermediate distributions (temperatures for SMC–Temp/AIS/PT; checkpoints for SMC–Train). Total Steps $= L \times K$ for SMC/AIS/PT and $= S_{\mathrm{mcmc}}$ for MCMC.

| Experiment | Method | Parallel | # Dists $L$ | Rejuv./Dist. $K$ | Total Steps |
|---|---|---|---|---|---|
| Ackley (10D) | MCMC–Final | $N{=}10,000$ | — | — | **50** |
| | SMC–Temp | $N{=}10,000$ | **10** | **5** | **50** |
| | SMC–Train | $N{=}10,000$ | **10** | **5** | **50** |
| | AIS–Auto | $N{=}10,000$ | **10** | **5** | **50** |
| | PT–MALA | $N{=}10,000$ | **10** | **5** | **50** |
| Superconductor | MCMC–Final | $N{=}500$ | — | — | **250** |
| | SMC–Temp | $N{=}500$ | **50** | **5** | **250** |
| | SMC–Train | $N{=}500$ | **50** | **5** | **250** |
| | AIS–Auto | $N{=}500$ | **50** | **5** | **250** |
| | PT–MALA | $N{=}500$ | **50** | **5** | **250** |

### K.4 DATA & MODELS BY TASK

**Ackley (10D).** **Model:** MLP with layers `[1024, 512, 256]`, Tanh activations. **Training data:** synthetic coverage over $[-10, 10]^{10}$ with three components: (i) uniform "plateau," (ii) stratified radial shells spanning target $f$-levels, and (iii) a small ball near the origin for additional $f \approx 0$ mass; 3,601,000 total points. **Target:** regress $f(x)$. **Sampling kernels:** MALA for all five methods.

**Superconductor (Design-Bench).** **Dimensions:** $d{=}87$. **Model:** MLP with layers `[2048, 2048]`, ReLU activations. **Training data:** train on the full available dataset; no oracle fine-tuning. **Target:** regress the provided score $f(x)$ (higher is better). **Sampling kernels:** MALA for all five methods.

**Novelty & Diversity (Superconductor).** Distances are computed in standardized feature space using the input scaler fit on the training set. *Novelty* is the per-sample $\ell_1$ (Manhattan) distance to the nearest training point (scikit-learn `NearestNeighbors`, `metric=manhattan`); we report the median and IQR across samples. *Diversity* is the median and IQR of pairwise $\ell_1$ distances among generated samples, computed over all unordered pairs ($n(n-1)/2$ for $n$ samples). All summaries are reported as median [IQR].

### K.5 EXTENDED RESULTS

**Ackley (10D, $\downarrow$).** Under matched compute, **SMC–Train** achieves the best mean and best-of-set with non-overlapping CIs relative to all baselines (Table 8). Quantitatively, **SMC–Train** reduces the mean objective vs. MCMC–Final by **17.8%** ($16.22 \to 13.33$), vs. SMC–Temp by **18.3%** ($16.31 \to 13.33$), vs. AIS by **18.3%** ($16.31 \to 13.33$), and vs. PT by **32.9%** ($19.87 \to 13.33$). On best-of-set, **SMC–Train** improves over MCMC–Final by **56.9%** ($8.56 \to 3.69$), over SMC–Temp by **53.0%** ($7.86 \to 3.69$), over AIS by **58.1%** ($8.81 \to 3.69$), and over PT by **72.9%** ($13.62 \to 3.69$). The 95% CIs for **SMC–Train** ($13.33\,[12.09, 14.58]$) are disjoint from the tight ranges of the other methods ($\approx 16.16 - 16.35$), indicating consistent improvement across seeds.

**Superconductor ($\uparrow$).** In high-dimensional materials design, **SMC–Train** leads both on best-of-set and mean (Table 8). Mean reward increases by **102.4%** vs. MCMC–Final ($76.68 \to 155.2$), by **606.1%** vs. SMC–Temp ($21.98 \to 155.2$), by **535.5%** vs. AIS ($24.42 \to 155.2$), and by **497.6%** vs. PT ($25.97 \to 155.2$). Relative to the reference score ($185.0$), the *mean* reaches **83.9%** of the target, while the *best-of-set* ($318.4$) is **172.1%** of the reference (i.e., +72.1% over target). These gains come with wider uncertainty for SMC–Train (95% CI: $[105.6, 204.8]$), reflecting more aggressive exploration that can land very high-reward candidates.

**On novelty and diversity.** Table 8 shows that AIS and PT achieve the highest *novelty* (median $\sim$35.3–35.7) and *diversity* (median $\sim$25.4–26.1), while **SMC–Train** is moderate on these axes (novelty 20.86; diversity 16.60). However, these higher exploration metrics do not translate into better objective quality: both AIS and PT have substantially lower mean rewards (24–26) than **SMC–Train** (155.2). We observe that the methods with the highest novelty/diversity also *retain many low-quality samples*, inflating dispersion-based metrics without improving the objective. In contrast, **SMC–Train** balances exploration and exploitation: it traverses the space broadly enough to discover strong candidates (best-of-set 318.4) while concentrating mass to raise the *mean* reward. Thus, *more novelty/diversity does not necessarily imply better design quality* when a significant tail of poor samples is preserved.

**Takeaways.** (i) Under matched compute, **SMC–Train** consistently outperforms baselines on Ackley and Superconductor by large margins in both mean and best-of-set. (ii) For Superconductor, apparent exploration advantages (higher novelty/diversity) from AIS/PT coincide with *lower* objective quality—suggesting these methods over-emphasize exploration and retain weak samples. (iii) Reporting both mean and best-of-set, alongside novelty/diversity, is essential: together, they show that **SMC–Train** drives objective gains while maintaining reasonable exploration, rather than chasing dispersion alone.

## K.6 FIGURES

### K.6.1 ACKLEY

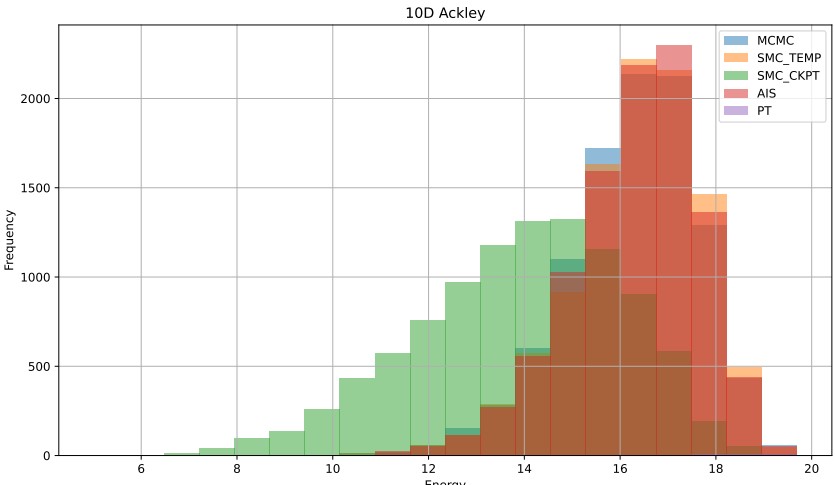

Figure 21: Full energy histogram for Ackley (10D). Samples from MCMC–Final, SMC–Temp, and **SMC–Train**.

### K.6.2 SUPERCONDUCTOR

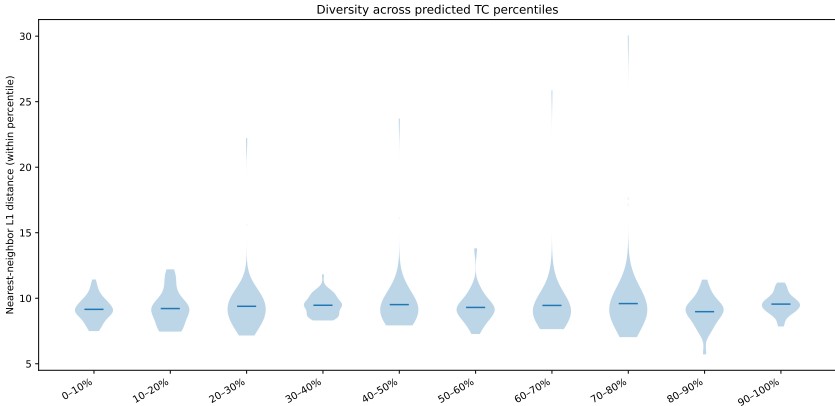

Figure 22: Samples are partitioned into Tc deciles (0–10%, ..., 90–100%). Within each bin, we plot the distribution of nearest-neighbor L1 distances (in standardized feature space) among the samples in that bin. Broad—and non-shrinking—within-bin L1 distributions at higher Tc percentiles indicate that sample diversity does not collapse as Tc increases.

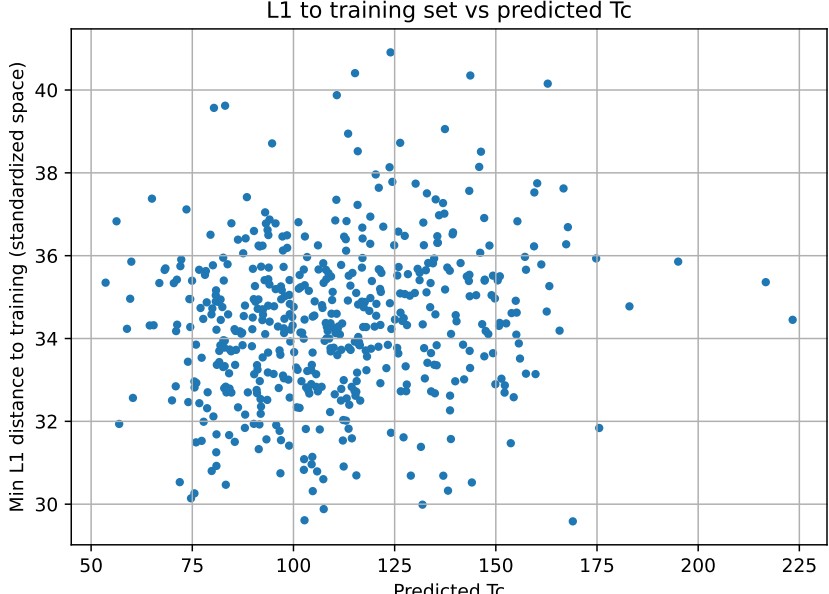

Figure 23: Each point shows a sample's predicted Tc (x-axis) versus its minimum L1 distance to any training example in standardized space (y-axis). High-Tc proposals do not systematically move closer to the training set; many top-Tc samples remain well separated, indicating genuine novelty rather than simple memorization.

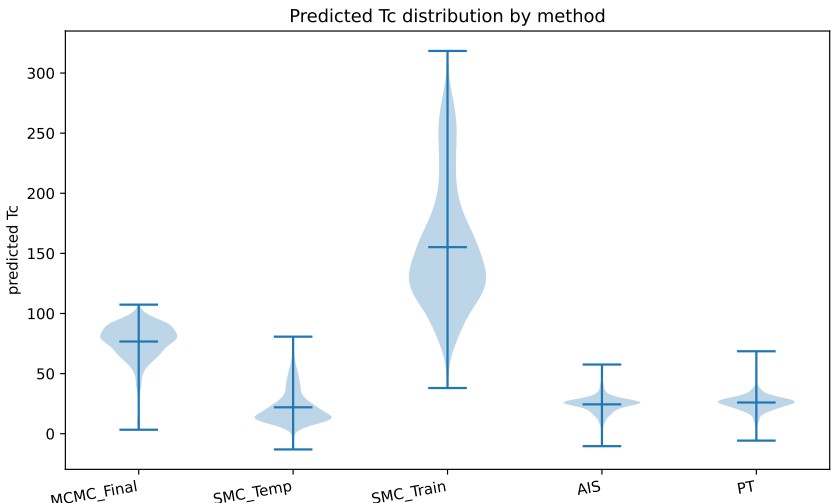

Figure 24: Side-by-side violins of the predicted critical temperature (Tc) for the three sampling methods, aggregated over seeds. The plot highlights differences in central tendency and tail behavior across methods.

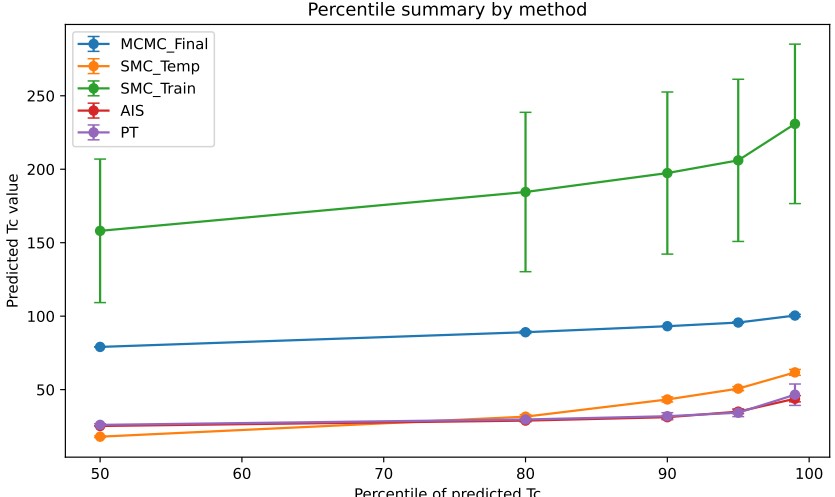

Figure 25: For each method, we compute selected Tc percentiles (50th–100th) per seed and report the mean ± standard deviation across seeds. Our method consistently gives higher Tc samples than the other two methods.

## L  USAGE OF LLMS

We utilize LLMs to assist with the writing of the paper. We provided GPT-5 an outline of our key points for each paragraph, and GPT-5 converted them to a paragraph format with latex formatting. We also utilized LLMs to research related work for each of our 4 sections.

