# OpenReview forum: "From Predictors to Samplers via the Training Trajectory"
_ICLR.cc/2026/Conference — ICLR 2026 Poster_

### Official Review · Reviewer_v2vM · 2025-10-30

**Soundness:** 4
**Presentation:** 3
**Contribution:** 4
**Rating:** 8
**Confidence:** 4

**Summary:**

### Summary

This paper leverages a well-celebrated theoretical idea, i.e. the coarse to fine, spectral learning dynamics of gradient based learning to help create a series of landscape with different smoothness. Then use these landscapes as annealed landscapes to help sampling from the final complex energy landscape. They conducted theory on simple boolean settings showing the learning order effect in the degree of polynomial, and then validated the idea in various discrete and continuous sampling set up and showed significant improvement upon simply temperature annealed sampling and other MC methods.

**Strengths:**

### Strength

- The paper is tackling an interesting problem with a very creative solution, connecting ideas from spectral bias, training dynamics and energy based model and sampling. I’m very convinced of the idea.
- The authors noted significant agreement with the theory in FCNN and MLP setting, and noted the deviation in transformer setting. We commend their honesty about the limitation of the theory.
- The experimental testing of the idea is very comprehensive and convincing, showing univocal benefit of this idea.

**Weaknesses:**

### Weakness

- Often it’s not clear which training time check point the authors used in sampling experiments.
    - e.g. in 4.1.2, it’s not clear from writing which training time checkpoint was used and what 1K and 10K steps denote. is it that 10K steps checkpoint is better intermediate landscape than 1K step?
    - More generally, I feel there is a lot of heuristics and design space for which checkpoint(s) are best suit for these intermediate landscapes, and how do the authors decide on them?

**Questions:**

- C.f. Sec. 3.2, the theory / motivation of the paper also aligns closely with the learning dynamics of score-based diffusion models, i.e. learned score vector fields are simpler, smooth earlier in the training, usually better approximated by a linear vector field. There is a nice spectral ordering for the learning of vector field and distribution. [^1], [^2]

[^1] Wang, & Vastola, (2024). The unreasonable effectiveness of gaussian score approximation for diffusion models and its applications. TMLR

[^2] Wang, & Pehlevan (2025). An analytical theory of power law spectral bias in the learning dynamics of diffusion models. NeurIPS

- For the idea of sampling leveraging a sequence of landscapes from smooth to rugged to help find tricky spiky solutions, the authors could also mention this recent work [^3], which shares a very similar picture, but did not use the learning dynamics to help. I feel [^3] could use many intuition / results from this paper to help learn their sequence of landscapes.

[^3] Du, Y., Mao, J., & Tenenbaum, J. B. (2024). Learning iterative reasoning through energy diffusion.

- In the continuous sampling experiments, is it relevant to compare to or report reference values for evolutionary algorithms? since these are also common functions used to benchmark those, e.g. CMAES.

- Discussing how to best heuristically pick the intermediate checkpoint is quite interesting and useful for the reader of the paper.

Minor

- The Table 2 format is a bit confusing….
the right column (step count) should not be compared on the same table with the mid and left column (success rate). A separatrix should be added or different table should be used to present the median step result.
From the first glance, it’s confusing to see 4.00 and 2.00 in the right column…. and median step is an integer so why do we have 2 decimal points here

- Table 5 and Sec. 4.1.2 are not very clear, which value is the temperature annealed GWG? why does it have two values as authors say they used the final checkpoint.

---

> ### Author Response · Authors · 2025-11-20
> **Reviewer Response**
>
> Thank you for your detailed review, along with suggestions for future directions. We are gratified to hear you find the idea very convincing, and the experiments very comprehensive.
>
> **Clarity Issues**
>
> Thank you for pointing out the clarity issues in the experiment tables; we will address those.
>
> **Checkpoint Selection**
>
> We use evenly spaced checkpoints for all experiments, and find that they generally work well. We also conduct an ablation on the # of checkpoints; please see 1.Checkpoint dependency in the general response. We find significant improvements from as little as 5 checkpoints.
>
> **Spectral ordering in diffusion models**
>
> This is a very interesting literature that we were not previously aware of. We will look more into this, and update the paper with these references. Thank you for also pointing out a connection to [3]; this is an interesting future direction for us.
>
> **Evolutionary algorithm comparison**
>
> We did not benchmark against evolutionary algorithms as we thought derivative-free methods would perform poorly, especially when the initial candidates are all poor. However, we will try to add a benchmark if we have time.

---

### Official Review · Reviewer_45Q3 · 2025-10-31

**Soundness:** 3
**Presentation:** 3
**Contribution:** 3
**Rating:** 6
**Confidence:** 2

**Summary:**

The paper proposes a method called trajectory annealing, which uses intermediate training checkpoints of a predictor to guide sampling. The main idea is that during training, neural networks naturally evolve from coarse to fine representations, early checkpoints smooth high-frequency variations, while later ones add detail. By running MCMC samplers (GWG for discrete and MALA for continuous variables) sequentially across these checkpoints, the method achieves better mixing and sampling efficiency, especially in rugged or synergistic landscapes. Experiments on Boolean, MNIST-EBM, DNA design, and materials datasets show strong empirical gains over standard temperature annealing. The theory part connects this to SGD’s hierarchical learning of low- to high-degree monomials.

**Strengths:**

The method is conceptually simple yet powerful, requires no retraining or architectural change, and leverages an inherent property of neural network learning. The connection to hierarchical degree learning and NTK smoothing is novel. Results are extensive and consistent across very different domains. The Boolean case study is particularly compelling, showing exponential-to-linear mixing improvements.

**Weaknesses:**

Theoretical results rely on strong assumptions (e.g., degree-wise alignment checkpoints) that may not generalize. Experiments are numerous but some seem cherry-picked to highlight advantages. Limited comparison with recent diffusion-based or amortized samplers. Some derivations could be more formal, and the transition from discrete to continuous domains feels hand-wavy. Also, claims of "no extra compute" ignore checkpoint storage and evaluation overhead.

**Questions:**

How sensitive is the performance to checkpoint spacing or number of steps per checkpoint? Could this be integrated with modern optimizer schedules or adaptive checkpoint selection? How does it perform on large-scale, non-convex tasks like ImageNet classifiers? Is there any insight on when the coarse-to-fine property breaks down, e.g., transformers?

---

> ### Author Response · Authors · 2025-11-20
> **Reviewer Response**
>
> We thank the reviewer for their detailed and thoughtful feedback. We are glad that you find our method powerful and conceptually novel.
>
> **Degree-wise alignment assumptions**
> We empirically test out this assumption under standard training in Figure 1, and provide further evidence in Appendix C across different widths, activations, layers, and architectures (e.g. CNN vs FCNN). We find that it appears to hold.
>
> **Experiment Choice**
> Our synthetic discrete experiments were specifically chosen to highlight findings from the theory. MNIST EBM was chosen as it is the largest discrete EBM model, and DNA design was chosen as the concept of a "motif" aligns with synergistic interactions. For the continuous experiments, Ackley is a standard non-convexity benchmark, and the materials design task was taken from a design benchmark.
>
> **Comparison with Diffusion**
> Please see 2. When to use our method vs. diffusion models in the general response. We are able to handle two scenarios - interpretability and constrained sampling - that diffusion models are not able to handle. Furthermore, if one is already training a predictor, simply sampling from it (using our method) is more compute-efficient than training a diffusion model from scratch.
>
> **Checkpoint Overhead/Choice**
> Please see the ablation in 1.Checkpoint dependency in the general response. Performing an ablation on our longest training experiment, we find significant improvements from even 5 checkpoints. The method is not sensitive to checkpoint spacing or steps per a checkpoint, for simplicity, we keep them evenly spaced and perform the same number of steps at each checkpoint.
>
> The only overhead is from memory to save the additional checkpoints. Inference time should not be affected as loading a 500M checkpoint from disk to GPU is <1 second. The user can decide how many checkpoints to save based on their memory constraints, and have those be evenly spaced.
>
> **Transformers**
> Our theory doesn't port over to transformers. In the discrete case, low degree monomials are only learned before high degree monomials because fewer variables leads to much higher gradient alignment. However, transformers have a completely different way of learning interactions (e.g. attention) where this does not carry over. Similarly, the NTK for FCNNs, CNNs, and ResNets have a bias towards lower-degree spherical harmonics, which aligns with the bias of gaussian smoothing. However, transformers have a different eigenfunction basis for their inductive bias.

---

### Official Review · Reviewer_wkfn · 2025-11-03

**Soundness:** 3
**Presentation:** 3
**Contribution:** 3
**Rating:** 6
**Confidence:** 3

**Summary:**

The paper proposes trajectory annealing: instead of sampling only from the final trained predictor $f^*$, the sampler runs short MCMC updates across saved training checkpoints $\{f_t\}$, exploiting the observed coarse‑to‑fine learning dynamics (early checkpoints damp high‑degree/high‑frequency components, later checkpoints restore detail).

**Strengths:**

- Leveraging the existing training trajectory as an annealing schedule is elegant and creative and requires no re‑training or auxiliary generative model. The coarse to fine picture is illustrated empirically and theoretically (Apps A-B).
- The method works for discrete (GWG) and continuous (SMC) domains.
- The Hamming‑ball constraint is handled naturally within the MCMC framework and yields large gains on DNA (Table 7).
- The $O(d\log d)$ mixing is theoretically supported on degree-1/2 surrogates.

**Weaknesses:**

- Compute parity (App J) is defined as matching the number of MALA steps, but SMC adds resampling/weight computations and multi‑checkpoint bookkeeping. MNIST also saves hundreds of checkpoints (50k training epochs saving every 100 steps). Reporting the wall‑clock time and memory would substantiate the “no additional compute” claim from the abstract.
- For the discrete synthetic tasks the only baseline is GWG (authors state this choice explicitly), leaving out stronger informed/non‑local samplers (e.g., locally balanced and discrete‑Langevin families). This narrows the comparison.
- The method does not apply to transformers (stated in the paper), which constrains scope for many modern applications.
- Several evaluations measure best‑of‑run (DNA keeps the best of 60 steps per run), which is an optimization metric. Mixing diagnostic or distributional metrics would be helpful.

**Questions:**

- With only a few steps at the final checkpoint, how biased are samples relative to the target distribution?
- How do performance and compute vary with the number and placement of checkpoints (uniform vs geometric; early‑only vs full trajectory)? MNIST uses 500 checkpoints, but discrete synthetic uses just a few.
- Have you tried saving checkpoints based on a performance metric instead of time?
- The paper states diffusion “cannot” handle Hamming‑ball constraints (p. 8). Could you qualify this?

---

> ### Author Response · Authors · 2025-11-20
> **Reviewer Response**
>
> We thank the reviewer for their detailed and thoughtful feedback. We are glad that you find our work elegant and creative.
>
> **Checkpoint Selection/No Additional Compute Claim**
>
> Thank you for raising this point. We have added an MNIST ablation (please look at 1.Checkpoint dependency in the general response), and find significant benefits with even 5 evenly spaced checkpoints.
>
> Across all experiments, our checkpoints are evenly spaced; we find that this works well. We haven't experimented with more sophisticated approaches (e.g. performance metrics, geometric, favoring early epochs).
>
> In practice, the user can decide in advance the # of checkpoints they want to save, based on their memory constraints. Loading a 500M model from disk onto a GPU takes <1 second, so the delta in wall-clock time should be very minimal.
>
> For the SMC compute, we are comparing against Temperature-Annealing SMC on just the final checkpoint. So the resampling/weight compute is the same across methods.
>
> **Discrete Synthetic Task Baselines**
>
> 1. *Conceptual Difference*
>
> All discrete samplers use the gradient to inform the proposal distribution. However, for functions with high-frequency, high-magnitude variation, the gradient is not useful - it will just steer you towards the local minima. If the temperature is relaxed, then one can escape the local minima but again, there is no guidance towards global minima.
>
> We bypass this issue by using the gradient from a previous checkpoint. Here, the high-frequency/high-magnitude variation is not present as the function is more coarse. Therefore, the gradient is more informative.
>
> We note that we are compatible with all samplers; any sampler can be run on checkpoints instead of on the final distribution.
>
> 2. *DLP Benchmark*
>
> We have benchmarked against the DLP sampler in the paper, A Langevin-like Sampler for Discrete Distributions, for our MNIST 1K experiment. We note similar improvements in FID score as the # of checkpoints increases.
>
> | Checkpoints | Mean FID (std)   |
> |-------------|------------------|
> | 0 (Baseline) | 24.56 (0.352)   |
> | 5           | 16.81 (0.235)    |
> | 10          | 15.66 (0.281)    |
> | 25          | 12.78 (0.272)    |
> | 50          | 11.84 (0.374)    |
> | 100         | 12.30 (0.396)    |
> | 500         | 12.23 (0.217)    |
>
> **Best-of-run Metrics**
> For our design tasks, we do not actually have good ground-truth distributions. Thus, we report best-of, diversity, and novelty metrics. We follow this reporting from existing design papers (e.g. Improving protein optimization with smoothed fitness landscapes).
>
> **Sample Bias From Final Checkpoint**
> Since we focus on rugged landscapes, the final checkpoint will largely guide a point to its local minima. We rely on earlier checkpoints for mixing/exploring the landscape.
>
> **Diffusion Cannot Handle Hamming-Ball Constraint**
> Please see the constrained sampling in section 2a. of our general response above. A predictor is naturally able to handle test-time constraints in a way that diffusion models are not. There have been workarounds in the diffusion literature, which utilize importance sampling from an auxiliary distribution (typically the constraint applied to the denoised prediction). However, this may require a prohibitive number of particles to estimate the distribution of interest.

---

### Official Review · Reviewer_zyoC · 2025-11-03

**Soundness:** 2
**Presentation:** 3
**Contribution:** 1
**Rating:** 2
**Confidence:** 4

**Summary:**

The paper proposes to use intermediate checkpoints from training a predictor model (used to define an energy function) to anneal MCMC from easy to sample distributions to the target distribution, which may have pathological structure that makes it difficult to directly sample from. They provide hierarchical learning analysis to validate the intuition, and provide some experiments to verify that the method works.

**Strengths:**

The idea of using the training checkpoints to perform some kind of annealing is novel to the best of my knowledge. I agree with their argument on how early stage checkpoints focus on high level features, late checkpoints focus on details, providing a sampling path.

The use of confidence intervals demonstrates thorough evaluation metrics.

The hierarchical learning analysis is also quite interesting.

**Weaknesses:**

**Literature Review**: The paper does not mention [2] or [3], which are focused on accessing modes that are difficult to reach. [3] is more recent so the omission is understandable.

**Base sampler**: Gibbs with gradient changes one coordinate at a time (at least in the default version), which makes it very slow. Are the benefits of annealing via model checkpoints preserved when using samplers proposed in [1, 2, 3]? Or does performance improvement from using this annealing strategy saturate with stronger base samplers? While GWG was chosen due to simplicity, it is important to note that more recent methods may solve this problem without the need for annealing across checkpoints.

**Breadth of Metrics**: It would be nice to include metrics that show sampling accuracy v.s number of sampling steps. For the MNIST experiment, it could take the form of log maximum mean divergence v.s sampling steps. Also, the paper does not include (Effective Sample Size), which is an important metric for evaluating the efficiency [1].

**Characterization of Diffusion**: They characterize diffusion as requiring training over the entire trajectory instead of single step MLE. However, I am not sure that this is a fair characterization: in practice, the score matching objective is trajectory free [4]. Even in discrete diffusion, the training objective takes the form of corrupting the input and then predicting the clean input [5, 6, 7, 8]. For obtaining SOTA results on CIFAR-10, it might take up to 15 hours. But it is entirely possible that within the first 15 minutes, the model is capable of generating reasonable images.

This is important because the intuition behind the proposed method is extremely similar to the core idea of diffusion: start from a distribution that is easy to sample, and gradually anneal it to the target distribution. The difference is that diffusion enables this with a single checkpoint, whereas the proposed method requires several checkpoints. Furthermore, diffusion directly supervises learning of the score (which is what GWG requires).

This is perhaps my largest concern with the submission: it seems to be capturing the intuition of diffusion, but via a more indirect path. If the focus of this paper is small models (as discussed in the introduction), is training a diffusion model really that expensive? And if the focus is on larger models where diffusion training is expensive, then it would also be expensive to store multiple checkpoints of the model and run analysis across all the checkpoints to determine which ones to use for annealing the sampler.

Also, I do not see a reference to [9], which directly incorporates the logic of diffusion into MCMC to improve mixing via the same intuition presented in this submission. While [9] is focused on the continuous domain, it may be worth considering how to extrapolate their method to the discrete space via gradient based discrete samplers.

[1] A Langevin-like Sampler for Discrete Distributions. Zhang et al. ICML 2022.

[2] Gradient-based Discrete Sampling with Automatic Cyclical Scheduling. Pynadath et al. NeurIPS 2024.

[3] Reheated Gradient-based Discrete Sampling for Combinatorial Optimization. Li, Zhang. TMLR 2025.

[4] Elucidating the Design Space of Diffusion-Based Generative Models. Karras et al. NeurIPS 2022.

[5] Simplified and Generalized Masked Diffusion for Discrete Data. Shi et al. NeurIPS 2024.

[6] Simple and Effective Masked Diffusion Language Models. Sahoo et al. NeurIPS 2024.

[7] Simple Guidance Mechanisms for Discrete Diffusion Models. Schiff et al. Preprint 2024.

[8] The Diffusion Duality. Sahoo et al. ICML 2025.

[9] Diffusive Gibbs Sampling. Chen et al. ICML 2024.

**Questions:**

- How does this method perform when using DLP sampler? Are the gains of using checkpoints / using just the final checkpoint just as large?
- In the introduction, it is stated that it takes 15 hours to train a diffusion model on CIFAR-10. Is there a citation for this? Are there plots of the loss v.s training time of diffusion methods for the tasks considered?

---

> ### Author Response · Authors · 2025-11-20
> **Reviewer Response**
>
> We thank the reviewer for their thoughtful and detailed feedback. Below, we address the raised points.
>
> **Other Discrete Samplers**
>
> *1.Conceptual Difference*
> All discrete samplers use the gradient to inform the proposal distribution. However, for functions with high-frequency, high-magnitude variation, the gradient is not useful - it will just steer you towards the local minima. If the temperature is relaxed, then one can escape the local minima but again, there is no guidance towards global minima.
>
> We bypass this issue by using the gradient from a previous checkpoint. Here, the high-frequency/high-magnitude variation is not present as the function is more coarse. Therefore, the gradient is more informative.
>
> [2] focuses on tuning the step size and balancing parameter, and [3] focuses on tuning the temperature. These are valuable contributions, but do not address our problem. We will add citations for [1,2,3] to our paper.
>
> *2. Q: Method Using DLP*
> We have rerun our MNIST experiment using the DLP sampler. We find that the numbers are similar to the above results (using the GWG sampler).
>
> | Checkpoints | Mean FID (std)   |
> |-------------|------------------|
> | 0 (Baseline) | 24.56 (0.352)   |
> | 5           | 16.81 (0.235)    |
> | 10          | 15.66 (0.281)    |
> | 25          | 12.78 (0.272)    |
> | 50          | 11.84 (0.374)    |
> | 100         | 12.30 (0.396)    |
> | 500         | 12.23 (0.217)    |
>
> 3. We note that we also show results on continuous domains, where Langevin MCMC along the trajectory outperforms Langevin MCMC on the final checkpoint.
>
> **Breadth of Metrics**
>
> 1. For MNIST, we show FID scores for 1K sampling steps and 10K sampling steps. We don’t do this for the other experiments as the number of steps is comparatively fewer.
>
> 2. [1] from your citation uses ESS as a single-chain mixing diagnostic for sampling from Ising. None of our experiments focus on single-chain mixing; each chain contributes only one sample.
>
> **Characterization of Diffusion**
>
> 1. *Conceptual similarity to Diffusion* Yes, our method works for the same reason that diffusion does. We make this similarity precise in Appendix F. Diffusion suppresses spherical harmonic of degree k at the rate $e^{-k^2}$. Under the NTK assumption, the training trajectory suppresses spherical harmonic of degree k at a rate that depends on the activation, $O\\left(k^{-d} e^{-\sqrt{k}}\right)$ for Tanh.
>
> 2. *When to use our method*. Please see 2 in the general response. Our method can assist with interpretability and constrained sampling, which diffusion models struggle with. Even 2-5 evenly spaced checkpoints yield large sampling gains, so it can easily work with large predictors. We don’t run any analysis to determine which checkpoints to use; given a number of checkpoints, we just pick evenly spaced ones.
>
> 3. [9]’s method noises an input and denoises by following gradient descent on the energy. However, for energy landscapes with high-magnitude, high-frequency variation (e.g. Ackley), the gradient is not particularly useful - it just steers you towards the local minima. We will add a citation for [9].
>
> 4. *Q: Training Time Citation* In the repo [1], they say training CIFAR-10 takes 2 days across 8 V-100 GPUs. They are aiming for SOTA results. In the repo [2] focused on a minimal implementation, they say training a “basic 35.7M UNet on CIFAR-10 takes ~14 hours” across 4 GPUs. Given that diffusion models are stochastic and do not use maximum likelihood, it makes sense that they take an order of magnitude longer to train, compared to a predictor.  We will add a citation to the paper.
>
> [1] https://github.com/NVlabs/edm
> [2] https://github.com/zhaisf/cDiffusion-cifar10

---

> > ### Comment · Reviewer_zyoC · 2025-11-20
> > **Response**
> >
> > I thank the authors for their rebuttal. I address some of the points below.
> >
> > **Other Discrete Samplers**
> >
> > The works I bring up are specifically designed to address the issue of gradients pointing towards local modes -- specifically, [1] uses a cyclical step-size to enable exploration and exploitation, and [2] uses annealing. Both methods alternate between using relaxed proposals that enable exploration to discover modes, and then leveraging the gradient to guide to the local optimum within the modes. These methods don't require any additional checkpoints, just control over the discrete sampler hyper-parameters.
> >
> > Furthermore, the point on gradients not being useful seems contradictory -- if they are not useful (as mentioned when discussing "Diffusive Gibbs Sampling" [3]), then why use gibbs with gradient? I believe the core point being made is that gradients are useful for exploitation within a mode but often hinder exploration between modes. However, [1, 2, 3] demonstrate how to counteract this by using either a cyclical step-size schedule, temperature annealing, or diffusion-like properties.
> >
> > I acknowledge the additional results for DLP strengthen the empirical case for the proposed method.
> >
> > **Breadth of Metrics**
> >
> > This point is no longer a concern, I thank the authors for their explanation.
> >
> > **Characterization of Diffusion**
> >
> > In the general response, the authors point to inference-time efficiency as a benefit of their method. In their rebuttal, they specifically point to the training times for diffusion models on CIFAR-10 as being costly. However, I still find this claimed advantage to be problematic.
> >
> > First, the training times are for CIFAR-10, which is not discussed in the submission. It does not make sense to claim an efficiency advantage over diffusion models based on CIFAR-10 training times when the submission presents no CIFAR-10 results whatsoever. **This is not a comparison - it is citing one number in isolation**. The authors cite 15 minutes training for a predictor -- but what is the source for this? Where can I find this result? Can this predictor generate actual images with similar quality to diffusion? The submission contains no results or citations supporting this. To demonstrate efficiency, the proposed method must also be trained on CIFAR-10 to establish that it is actually cheaper. Otherwise, the efficiency claim is simply not factual.
> >
> > I find the efficiency claim particularly difficult to believe since training EBMs (the model used for the MNIST experiments) are fairly expensive. For example, we can look to [4] that trains the CIFAR EBM for 2 days on a single GPU. [4] is particularly relevant as it establishes the modern deep EBM training approach used in [5] (the submission which uses the code-base from [5]). [5] explicitly uses the replay buffer approach from [4] when training deep EBMs (see Section 8 in [5]).
> >
> > This shows that training an EBM (which is what they use in the MNIST experiment) and a diffusion model have roughly similar training times, contradicting the efficiency claim. In regards to the minimal diffusion model that is trained for 14 hours on 4 V100 GPUs, it would be possible to train on a single GPU via gradient accumulation / reducing batch size.  This should result in a 56 (4 * 14) hour run, roughly matching the 2 day time budget for training an EBM.
> >
> > Furthermore, it is crucial to note that the 48 hour training cost was for a 5 million parameter EBM, whereas the [6] is training a 45 million parameter diffusion model -- it is quite likely that reducing the diffusion parameter count would substantially decrease the 56 hour training time (which is already close to the 48 hour budget for EBMs). Lastly, the diffusion model outperforms the EBM in terms of FID (3.54 v.s 40.58). Since a smaller EBM takes almost the same amount of time to train as a diffusion model and performs worse, I find the efficiency argument to be inaccurate.
> >
> > Overall, I appreciate the papers exploration of an interesting direction, and the constrained sampling experiments do show some promise. However, this paper seems to be solving a problem that other discrete gradient methods [1,2,3] solve without storing multiple checkpoints, and the discussion of these related works does not illuminate any advantage of the proposed method. Additionally, my concerns about the characterization of diffusion remain unresolved. Thus I maintain my score.
> >
> > [1] Gradient-based Discrete Sampling with Automatic Cyclical Scheduling. Pynadath et al. NeurIPS 2024.
> >
> > [2] Reheated Gradient-based Discrete Sampling for Combinatorial Optimization. Li, Zhang. TMLR 2025.
> >
> > [3] Diffusive Gibbs Sampling. Chen et al. ICML 2024
> >
> > [4] Implicit Generation and Modeling with Energy-Based Models. Yilun Du, Igor Mordatch. NeurIPS 2020.
> >
> > [5] Oops I Took A Gradient: Scalable Sampling for Discrete Distributions. Grathwohl et al. ICML 2021.
> >
> > [6] https://github.com/zhaisf/cDiffusion-cifar10

---

> ### Author Response · Authors · 2025-11-23
> **Reviewer Response (1/2)**
>
> Thank you for your extensive comments. Your feedback has been very helpful in making the paper’s claims more precise/clear.
>
> **Other Discrete Samplers**
>
> We would like to use a simple toy example to highlight the effect of adding trajectory-based checkpoints on top of existing samplers. Let $f(x) = 0.1 \sum_{i=0}^{8} x_i + 3.0 \prod_{i=0}^{8} x_i$. $x$ also contains other spurious variables that don’t contribute to the output.
>
> We would like to find the global max of this function, which is all 1s. However, it is dominated by high-frequency maxima. If the product of the first 8 terms is -1, the gradient for each bit will suggest flipping, and if the product is 1, then the gradient for each bit will suggest not flipping (staying the same).
>
> Given a random starting point, we will immediately end up in a local maxima. To escape this local maxima, ACS increases the step size alpha to make bigger moves, decreases beta to de-prioritize gradient direction, and ResCO increases the temperature. Crucially, all of these introduce randomness into the proposal - which allows us to escape the local maxima. However, this new point is effectively a random choice among basins: the proposal that took us there offers no useful directional signal toward the global maximum, so the search between basins behaves essentially like a random walk.
>
> Now, consider our method. At an earlier checkpoint, the learned function is just $0.1 \sum_{i=0}^{8} x_i$. Because the high-frequency variation is absent, a single gradient step allows us concentrate the chain near the true global optimum.
>
> The broader point is that when the high-frequency variation is absent, in the earlier checkpoints, it’s very easy to discover high probability regions. The purpose of the final checkpoint is just to refine the samples. This is the same intuition behind diffusion, as you pointed out earlier.
>
> However, adjusting hyperparameters (e.g. temperature) cannot give us the same effect for pathological functions dominated by high-frequency, high-magnitude local variation. Tuning can only shake the chain out of a local maximum via essentially undirected noise. The resulting moves provide no useful directional signal toward the global maxima.
>
> To make this point more clear, we have benchmarked our method against the discrete samplers you listed on the toy function from above. Across 200 random particles, we measure the % that a chain starting at each particle, at any point in time, hits the global max. Our method, run for 2 steps on checkpoint 25 and 2 steps on the final checkpoint, is able to consistently find the global max in 4 steps with the DLP sampler. However, other methods run for 200 steps, on only the final checkpoint, are able to find the global max less consistently.
>
> | Method                               | Estimate | 95% CI             |
> |--------------------------------------|----------|----------------------|
> | Our Method (4 steps of DMALA)        | 0.9200   | [0.8816, 0.9584]     |
> | DMALA (200 steps)                    | 0.2800   | [0.2165, 0.3435]     |
> | ACS (200 steps, 20 tuning steps)     | 0.5900   | [0.5204, 0.6596]     |
> | ResCO (200 steps)                    | 0.1400   | [0.0909, 0.1891]     |
>
> The same argument applies to the sampler [9] that you mentioned. They apply noise to the input, and denoise via following the gradient of the energy. For functions dominated by high-magnitude, high-frequency variation, neither of these steps take us towards the global maxima.
>
> Lastly, we view our method as complementary to these samplers: they can serve as the base MCMC kernel, while our contribution is to run them along the training trajectory instead of only at the final checkpoint. This offers the unique benefit of changing the underlying energy landscape to be smoother.

---

> > ### Author Response · Authors · 2025-11-23
> > **Reviewer Response (2/2)**
> >
> > **Training time / comparison to diffusion**
> >
> > We appreciate the reviewer’s detailed comments and agree that our current discussion of diffusion training time is confusing and not well supported.
> >
> > 1. *Scope of our claims.*
> >    We do not claim that our method produces better samples than diffusion models, nor that it is more efficient than training a state-of-the-art diffusion model on the same dataset. If one is willing and able to train a high-quality diffusion model, diffusion will generally achieve lower FID than sampling from a predictor with our method.
> >
> >    Our intended use case is different:
> >
> >    **a.** scenarios where diffusion models are not directly applicable (see 2a in the general response), and
> >
> >    **b.** scenarios where a large predictor has already been trained for a prediction task, and the practitioner is in a compute-constrained regime where training an additional generative model is not feasible. In this setting, reusing the existing predictor with our trajectory-annealed MCMC requires only test-time sampling compute, whereas training a diffusion (or any new generative) model would require substantial extra training compute.
> >
> > 2. *Removal of CIFAR-10 timing claims.*
> >    The reviewer is correct that our CIFAR-10 training-time numbers (and the “15 minutes vs 15 hours” comparison) are out of place, since we do not present CIFAR-10 experiments in this submission and did not provide a controlled comparison.
> >
> > 3. *EBMs vs diffusion.*
> >    We agree that EBMs are very inefficient to learn (more inefficient than diffusion models). The EBM experiment was simply included for variety.
> >
> > We will revise the paper to state this more clearly and to frame diffusion as a complementary alternative rather than an efficiency baseline. We will remove all numerical training-time claims and dataset-specific comparisons to diffusion from the revised version and restrict ourselves to the qualitative point that our method does not require training any additional model beyond the predictor. We thank the reviewer for making this point.

---

### Author Response · Authors · 2025-11-19
**General Response**

We sincerely appreciate the reviewers’ time, effort, and constructive engagement. We are gratified that all reviewers agree with our core point: neural networks learn in a coarse-to-fine manner and this trajectory is useful for accelerating sampling. We are also glad to hear the reviewers found our experiments to be comprehensive and consistent.

The reviewers’ main questions focus on 1) Checkpoint dependency, and 2) What scenarios to use our method vs. diffusion models.

We address both below.

**1.Checkpoint dependency**

Our checkpoint selection tuning is minimal. Multiple checkpoints are always evenly spaced.
For all discrete experiments that train quickly, we just use epoch 25. We also add epoch 50 for the DNA experiment as that takes longer to train.

Our largest experiment is the MNIST EBM which is trained for 50,000 epochs. Thus, we conduct an ablation on the number of checkpoints. For a given # of checkpoints, the checkpoints are equally spaced across the training epochs.

As you can see below, we see dramatic gains from just 5 checkpoints, where more checkpoints leads to diminishing marginal returns.

The key empirical observation is that the trajectory monotonically evolves from coarse to fine. Therefore, any set of evenly spaced epochs leverages this property and speeds up sampling.

**MNIST Ablation**

| Checkpoints | Mean FID (std) |
| --- | --- |
| 0 (Baseline) | 29.61 (0.239) |
| 5 | 16.10 (0.237) |
| 10 | 14.42 (0.322) |
| 25 | 12.56 (0.289) |
| 50 | 11.93 (0.355) |
| 100 | 12.08 (0.435) |
| 500 | 11.73 (0.284) |


**2. When to use our method vs. diffusion models**

We separate our answer into (a) functionality that diffusion models fundamentally do not offer in this setting, and (b) compute efficiency in realistic workflows.

*2a. What we do that diffusion models can’t do*

*Interpretability*

Before deploying a trained predictor, we need to audit the function it has learned. This requires sampling from that predictor. For instance, we may want to:
- Sample minimal counterfactual edits to understand what’s driving a classification.
- Sample inputs (under domain constraints) that lead to outlier predictions.
- Sample high-scoring inputs that violate domain constraints, to surface failure modes.

Our method directly improves sampling from a trained predictor $f^*$, addressing what interpretability workflows need.
By contrast, diffusion models are separate generative models that are not related to the predictor.

*Easy Constrained Sampling*

Suppose we want to sample from the distribution of high-fitness variants conditional on a hard constraint, e.g., that $x$ lies within Hamming distance $\leq k$ of a given seed $x_0$. With a trained predictor / energy model $f^\star$, this is straightforward: we define the constrained target
\begin{equation}
    \pi_C(x) \propto \{f^\star(x)\}\\mathbf{1}_{\{d(x,x_0)\leq k\}}
\end{equation}
and run MCMC that simply rejects proposals leaving the constraint set. In other words, new test-time constraints are incorporated by modifying the energy or the acceptance rule; no retraining is required.

For diffusion models there is no equally simple mechanism that yields exact samples from $p(x \mid C)$ using a single pre-trained denoiser. Training-free guidance methods approximate the conditional score by differentiating a loss at the denoised prediction $\hat{x}_0(x_t)$, rather than the intractable expectation over $x_0$; as shown in Eqs.~(4-6) of [1], this approximation changes the target distribution and does not implement the desired hard constraint. To encode constraints of the form $d(x,x_0)\leq k$ in a principled way, one typically has to train an auxiliary guidance network for each constraint.

[1] Understanding and Improving Training-free Loss-based Diffusion Guidance. NeurIPS 2024.

A proposed solution is the Twisted Diffusion Sampler (TDS) [2], which wraps the diffusion model in a Sequential Monte Carlo (SMC) scheme. TDS simulates many reverse-diffusion trajectories (particles) under a twisted proposal built from heuristic guidance, and applies importance weights and resampling at each step so that, as the number of particles $N \to \infty$, the empirical particle distribution converges to the true conditional. This comes at a computational cost: we need to run the reverse process $N$ times, and sharp or rare constraints lead to highly imbalanced importance weights unless large particle sets are used.

[2] Practical and Asymptotically Exact Conditional Sampling in Diffusion Models. NeurIPS 2023.

By contrast, once $f^\star$ is trained, adding a new hard constraint $C(x)$ in our predictor-based sampler is trivial: we simply restrict the MCMC chain to $C$ by modifying the acceptance rule. This plug-and-play ability to impose arbitrary constraints at test time, without additional training or heavy SMC machinery, is a practical advantage of standard energy-based models over current diffusion-based pipelines in our setting.

---

> ### Author Response · Authors · 2025-11-19
> **General Response (Cont.)**
>
> *2b. Where we are more compute-efficient than diffusion models*
>
> Suppose someone has just trained a large predictor on a fitness task. The next step, very naturally, is to use the predictor to mutate existing sequences to improve fitness. This is standard practice in the literature.
>
> We are able to make this more efficient. According to the ablation in 1), saving as few as 2-5, evenly-spaced, checkpoints throughout training can dramatically boost test-time sampling efficiency. This is simple to do, and makes sense from a cost-benefit standpoint.
>
> An alternate approach could be to then train a diffusion model, along with classifier conditioned guidance. However, this is effort and compute-intensive.

---

### Meta-Review · Area_Chair_PdNZ · 2026-01-08

**Summary:**

the paper addresses issues related to sampling from trained predictors. For addressing these issues,
they propose a method called trajectory annealing, which uses intermediate training checkpoints of a predictor to guide sampling.
The principal idea is that during training, neural networks naturally evolve from coarse
to fine representations and thus early checkpoints smooth high-frequency variations, while later ones add detail.
The authors proposes a method that runs on MCMC samplers sequentially across these checkpoints.

Most reviewers are supportive of the paper and find the idea of using training trajectory compelling. In addition,
they have found the connection with NTK novel and that the model performs well accross different domains (discrete and
Continuous). Based on al the positive feedbacks, I propose to recommend acceptance.

**Reviewer Concerns:**

adressed:
- metrics
- discrete samplers

**Reviewer Scores:**

I am not able to answer this question

---

### Decision · Program_Chairs · 2026-01-26

Accept (Poster)